# Bayesian Few-Shot Classification with One-vs-Each Pólya-Gamma Augmented Gaussian Processes

**Jake C. Snell**
University of Toronto
Vector Institute
jsnell@cs.toronto.edu

**Richard Zemel**
University of Toronto
Vector Institute
Canadian Institute for Advanced Research
zemel@cs.toronto.edu

## ABSTRACT

Few-shot classification (FSC), the task of adapting a classifier to unseen classes given a small labeled dataset, is an important step on the path toward human-like machine learning. Bayesian methods are well-suited to tackling the fundamental issue of overfitting in the few-shot scenario because they allow practitioners to specify prior beliefs and update those beliefs in light of observed data. Contemporary approaches to Bayesian few-shot classification maintain a posterior distribution over model parameters, which is slow and requires storage that scales with model size. Instead, we propose a Gaussian process classifier based on a novel combination of Pólya-Gamma augmentation and the one-vs-each softmax approximation (Titsias, 2016) that allows us to efficiently marginalize over functions rather than model parameters. We demonstrate improved accuracy and uncertainty quantification on both standard few-shot classification benchmarks and few-shot domain transfer tasks.

## 1 INTRODUCTION

Few-shot classification (FSC) is a rapidly growing area of machine learning that seeks to build classifiers able to adapt to novel classes given only a few labeled examples. It is an important step towards machine learning systems that can successfully handle challenging situations such as personalization, rare classes, and time-varying distribution shift. The shortage of labeled data in FSC leads to uncertainty over the parameters of the model, known as *model uncertainty* or *epistemic uncertainty*. If model uncertainty is not handled properly in the few-shot setting, there is a significant risk of overfitting. In addition, FSC is increasingly being used for risk-averse applications such as medical diagnosis (Prabhu, 2019) and human-computer interfaces (Wang et al., 2019) where it is important for a few-shot classifier to know when it is uncertain.

Bayesian methods maintain a distribution over model parameters and thus provide a natural framework for capturing this inherent model uncertainty. In a Bayesian approach, a prior distribution is first placed over the parameters of a model. After data is observed, the posterior distribution over parameters is computed using Bayesian inference. This elegant treatment of model uncertainty has led to a surge of interest in Bayesian approaches to FSC that infer a posterior distribution over the weights of a neural network (Finn et al., 2018; Yoon et al., 2018; Ravi & Beatson, 2019).

Although conceptually appealing, there are several practical obstacles to applying Bayesian inference directly to the weights of a neural network. Bayesian neural networks (BNNs) are expensive from both a computational and memory perspective. Moreover, specifying meaningful priors in parameter space is known to be difficult due to the complex relationship between weights and network outputs (Sun et al., 2019).

Gaussian processes (GPs) instead maintain a distribution over *functions* rather than model parameters. The prior is directly specified by a mean and covariance function, which may be parameterized by deep neural networks. When used with Gaussian likelihoods, GPs admit closed form expressions for the posterior and predictive distributions. They exchange the computational drawbacks of BNNs

for cubic scaling with the number of examples. In FSC, where the number of examples is small, this is often an acceptable trade-off.

When applying GPs to classification with a softmax likelihood, the non-conjugacy of the GP prior renders posterior inference intractable. Many approximate inference methods have been proposed to circumvent this, including variational inference and expectation propagation. In this paper we investigate a particularly promising class of approaches that augment the GP model with a set of auxiliary random variables, such that when they are marginalized out the original model is recovered (Albert & Chib, 1993; Girolami & Rogers, 2006; Linderman et al., 2015). Such augmentation-based approaches typically admit efficient Gibbs sampling procedures for generating posterior samples which when combined with Fisher's identity (Douc et al., 2014) can be used to optimize the parameters of the mean and covariance functions.

In particular, augmentation with Pólya-Gamma random variables (Polson et al., 2013) makes inference tractable in logistic models. Naively, this is useful for handling binary classification, but in this paper we show how to extend Pólya-Gamma augmentation to multiple classes by using the one-vs-each softmax approximation (Titsias, 2016), which can be expressed as a product of logistic sigmoids. We further show that the one-vs-each approximation can be interpreted as a composite likelihood (Lindsay, 1988; Varin et al., 2011), a connection which to our knowledge has not been made in the literature.

In this work, we make several contributions:

- We show how the one-vs-each softmax approximation (Titsias, 2016) can be interpreted as a composite likelihood consisting of pairwise conditional terms.
- We propose a novel GP classification method that combines the one-vs-each softmax approximation with Pólya-Gamma augmentation for tractable inference.
- We demonstrate competitive classification accuracy of our method on standard FSC benchmarks and challenging domain transfer settings.
- We propose several new benchmarks for uncertainty quantification in FSC, including calibration, robustness to input noise, and out-of-episode detection.
- We demonstrate improved uncertainty quantification of our method on the proposed benchmarks relative to standard few-shot baselines.

## 2 RELATED WORK

Our work is related to both GP methods for handling non-conjugate classification likelihoods and Bayesian approaches to few-shot classification. We summarize relevant work here.

### 2.1 GP CLASSIFICATION

**Non-augmentation approaches.** There are several classes of approaches for applying Gaussian processes to classification. The most straightforward method, known as least squares classification (Rifkin & Klautau, 2004), treats class labels as real-valued observations and performs inference with a Gaussian likelihood. The Laplace approximation (Williams & Barber, 1998) constructs a Gaussian approximate posterior centered at the posterior mode. Variational approaches (Titsias, 2009; Matthews et al., 2016) maximize a lower bound on the log marginal likelihood. In expectation propagation (Minka, 2001; Kim & Ghahramani, 2006; Hernandez-Lobato & Hernandez-Lobato, 2016), local Gaussian approximations to the likelihood are fitted iteratively to minimize KL divergence from the true posterior.

**Augmentation approaches.** Augmentation-based approaches introduce auxiliary random variables such that the original model is recovered when marginalized out. Girolami & Rogers (2006) propose a Gaussian augmentation for multinomial probit regression. Linderman et al. (2015) utilize Pólya-Gamma augmentation (Polson et al., 2013) and a stick-breaking construction to decompose a multinomial distribution into a product of binomials. Galy-Fajou et al. (2020) propose a logistic-softmax likelihood for classification and uses Gamma and Poisson augmentation in addition to Pólya-Gamma augmentation in order to perform inference.

## 2.2 FEW-SHOT CLASSIFICATION

**Meta-learning.** A common approach to FSC is meta-learning, which seeks to learn a strategy to update neural network parameters when faced with a novel learning task. The Meta-learner LSTM (Ravi & Larochelle, 2017) learns a meta-level LSTM to recurrently output a new set of parameters for a base learner. MAML (Finn et al., 2017) learns initializations of deep neural networks that perform well on task-specific losses after one or a few steps of gradient descent by backpropagating through the gradient descent procedure itself. LEO (Rusu et al., 2019) performs meta-learning in a learned low-dimensional latent space from which the parameters of a classifier are generated.

**Metric learning.** Metric learning approaches learn distances such that input examples can be meaningfully compared. Siamese Networks (Koch, 2015) learn a shared embedding network along with a distance layer for computing the probability that two examples belong to the same class. Matching Networks (Vinyals et al., 2016) uses a nonparametric classification in the form of attention over nearby examples, which can be interpreted as a form of soft $k$-nearest neighbors in the embedding space. Prototypical Networks (Snell et al., 2017) make predictions based on distances to nearest class centroids. Relation Networks (Sung et al., 2018) instead learn a more complex neural network distance function on top of the embedding layer.

**Bayesian Few-shot Classification.** More recently, Bayesian FSC approaches that attempt to infer a posterior over task-specific parameters have appeared. Grant et al. (2018) reinterpret MAML as an approximate empirical Bayes algorithm and propose LLAMA, which optimizes the Laplace approximation to the marginal likelihood. Bayesian MAML (Yoon et al., 2018) instead uses Stein Variational Gradient Descent (SVGD) (Liu & Wang, 2016) to approximate the posterior distribution over model parameters. VERSA (Gordon et al., 2019) uses amortized inference networks to obtain an approximate posterior distribution over task-specific parameters. ABML (Ravi & Beatson, 2019) uses a few steps of Bayes by Backprop (Blundell et al., 2015) on the support set to produce an approximate posterior over network parameters. CNAPs (Requeima et al., 2019) modulate task-specific Feature-wise Linear Modulation (FiLM) (Perez et al., 2018) layer parameters as the output of an adaptation network that takes the support set as input.

**GPs for Few-shot Learning.** There have been relatively few works applying GPs to few-shot learning. Tossou et al. (2020) consider Gaussian processes in the context of few-shot regression with Gaussian likelihoods. Deep Kernel Transfer (DKT) (Patacchiola et al., 2020) uses Gaussian processes with least squares classification to perform few-shot classification and learns covariance functions parameterized by deep neural networks. More recently, Titsias et al. (2020) applies GPs to meta-learning by maximizing the mutual information between the query set and a latent representation of the support set.

## 3 BACKGROUND

In this section we first review Pólya-Gamma augmentation for binary classification and the one-vs-each approximation before we introduce our method in Section 4.

### 3.1 PÓLYA-GAMMA AUGMENTATION

The Pólya-Gamma augmentation scheme was originally introduced to address Bayesian inference in logistic models (Polson et al., 2013). Suppose we have a vector of logits $\boldsymbol{\psi} \in \mathbb{R}^N$ with corresponding binary labels $\mathbf{y} \in \{0, 1\}^N$. The logistic likelihood is

$$p(\mathbf{y}|\boldsymbol{\psi}) = \prod_{i=1}^{N} \sigma(\psi_i)^{y_i}(1 - \sigma(\psi_i))^{1-y_i} = \prod_{i=1}^{N} \frac{(e^{\psi_i})^{y_i}}{1 + e^{\psi_i}}, \tag{1}$$

where $\sigma(\cdot)$ is the logistic sigmoid function. Let the prior over $\boldsymbol{\psi}$ be Gaussian: $p(\boldsymbol{\psi}) = \mathcal{N}(\boldsymbol{\psi}|\boldsymbol{\mu}, \boldsymbol{\Sigma})$. In Bayesian inference, we are interested in the posterior $p(\boldsymbol{\psi}|\mathbf{y}) \propto p(\mathbf{y}|\boldsymbol{\psi})p(\boldsymbol{\psi})$ but the form of (1) does not admit analytic computation of the posterior due to non-conjugacy. The main idea of Pólya-Gamma augmentation is to introduce auxiliary random variables $\boldsymbol{\omega}$ to the likelihood such that the original model is recovered when $\boldsymbol{\omega}$ is marginalized out: $p(\mathbf{y}|\boldsymbol{\psi}) = \int p(\boldsymbol{\omega})p(\mathbf{y}|\boldsymbol{\psi}, \boldsymbol{\omega}) \, d\boldsymbol{\omega}$.

Conditioned on $\boldsymbol{\omega} \sim \text{PG}(1,0)$, the batch likelihood is proportional to a diagonal Gaussian (see Section A for a full derivation):

$$p(\mathbf{y}|\boldsymbol{\psi}, \boldsymbol{\omega}) \propto \prod_{i=1}^{N} e^{-\omega_i \psi_i^2/2} e^{\kappa_i \psi_i} \propto \mathcal{N}(\boldsymbol{\Omega}^{-1} \boldsymbol{\kappa} \,|\, \boldsymbol{\psi}, \boldsymbol{\Omega}^{-1}), \tag{2}$$

where $\kappa_i = y_i - 1/2$ and $\boldsymbol{\Omega} = \text{diag}(\boldsymbol{\omega})$. The conditional distribution over $\boldsymbol{\psi}$ given $\mathbf{y}$ and $\boldsymbol{\omega}$ is now tractable:

$$p(\boldsymbol{\psi}|\mathbf{y}, \boldsymbol{\omega}) \propto p(\mathbf{y}|\boldsymbol{\psi}, \boldsymbol{\omega})p(\boldsymbol{\psi}) \propto \mathcal{N}(\boldsymbol{\psi}|\tilde{\boldsymbol{\Sigma}}(\boldsymbol{\Sigma}^{-1}\boldsymbol{\mu} + \boldsymbol{\kappa}), \tilde{\boldsymbol{\Sigma}}), \tag{3}$$

where $\tilde{\boldsymbol{\Sigma}} = (\boldsymbol{\Sigma}^{-1} + \boldsymbol{\Omega})^{-1}$. The conditional distribution of $\boldsymbol{\omega}$ given $\boldsymbol{\psi}$ and $\mathbf{y}$ can also be easily computed:

$$p(\omega_i|y_i, \psi_i) \propto \text{PG}(\omega_i|1,0)e^{-\omega_i \psi_i^2/2} \propto \text{PG}(\omega_i|1, \psi_i), \tag{4}$$

where the last expression follows from the exponential tilting property of Pólya-Gamma random variables. This suggest a Gibbs sampling procedure in which iterates $\boldsymbol{\omega}^{(t)} \sim p(\boldsymbol{\omega}|\mathbf{y}, \boldsymbol{\psi}^{(t-1)})$ and $\boldsymbol{\psi}^{(t)} \sim p(\boldsymbol{\psi}|\mathbf{X}, \mathbf{y}, \boldsymbol{\omega}^{(t)})$ are drawn sequentially until the Markov chain reaches its stationary distribution, which is the joint posterior $p(\boldsymbol{\psi}, \boldsymbol{\omega}|\mathbf{y})$. Fortunately, efficient samplers for the Pólya-Gamma distribution have been developed (Windle et al., 2014) to facilitate this.

## 3.2 One-vs-Each Approximation to Softmax

The one-vs-each (OVE) approximation (Titsias, 2016) was formulated as a lower bound to the softmax likelihood in order to handle classification over a large number of output classes, where computation of the normalizing constant is prohibitive. We employ the OVE approximation not to deal with extreme classification, but rather due to its compatibility with Pólya-Gamma augmentation, as we shall soon see. The one-vs-each approximation can be derived by first rewriting the softmax likelihood as follows:

$$p(y = i \,|\, \mathbf{f}) \triangleq \frac{e^{f_i}}{\sum_j e^{f_j}} = \frac{1}{1 + \sum_{j \neq i} e^{-(f_i - f_j)}}, \tag{5}$$

where $\mathbf{f} \triangleq (f_1, \ldots, f_C)^\top$ are the logits. Since in general $\prod_k (1 + \alpha_k) \geq (1 + \sum_k \alpha_k)$ for $\alpha_k \geq 0$, the softmax likelihood (5) can be bounded as follows:

$$p(y = i \,|\, \mathbf{f}) \geq \prod_{j \neq i} \frac{1}{1 + e^{-(f_i - f_j)}} = \prod_{j \neq i} \sigma(f_i - f_j), \tag{6}$$

which is the OVE lower bound. This expression avoids the normalizing constant and factorizes into a product of pairwise sigmoids, which is amenable to Pólya-Gamma augmentation for tractable inference.

## 4 One-vs-Each Pólya-Gamma GPs

In this section, we first show how the one-vs-each (OVE) approximation can be interpreted as a pairwise composite likelihood. We then we introduce our method for GP-based Bayesian few-shot classification, which brings together OVE and Pólya-Gamma augmentation in a novel combination.

### 4.1 OVE as a Composite Likelihood

Titsias (2016) showed that the OVE approximation shares the same global optimum as the softmax maximum likelihood, suggesting a close relationship between the two. We show here that in fact OVE can be interpreted as a pairwise *composite likelihood* version of the softmax. Composite likelihoods (Lindsay, 1988; Varin et al., 2011) are a type of approximate likelihood often employed when the exact likelihood is intractable or otherwise difficult to compute. Given a collection of marginal or conditional events $\{E_1, \ldots, E_K\}$ and parameters $\mathbf{f}$, a composite likelihood is defined as:

$$\mathcal{L}_{\text{CL}}(\mathbf{f} \,|\, y) \triangleq \prod_{k=1}^{K} \mathcal{L}_k(\mathbf{f} \,|\, y)^{w_k}, \tag{7}$$

where $\mathcal{L}_k(\mathbf{f}\,|\,y) \propto p(y \in E_k\,|\,\mathbf{f})$ and $w_k \geq 0$ are arbitrary weights.

In order to make the connection to OVE, it will be useful to let the one-hot encoding of the label $y$ be denoted as $\mathbf{y} \in \{0,1\}^C$. Define a set of $C(C-1)/2$ pairwise conditional events $E_{ij}$, one each for all pairs of classes $i \neq j$, indicating the event that the model's output matches the target label for classes $i$ and $j$ conditioned on all the other classes:

$$p(\mathbf{y} \in E_{ij}\,|\,\mathbf{f}) \triangleq p(y_i, y_j\,|\,\mathbf{y}_{\neg ij}, \mathbf{f}), \tag{8}$$

where $\neg ij$ denotes the set of classes not equal to either $i$ or $j$. This expression resembles the pseudolikelihood (Besag, 1975), but instead of a single conditional event per output site, the expression in (8) considers all pairs of sites. Stoehr & Friel (2015) explored similar composite likelihood generalizations of the pseudolikelihood in the context of random fields.

Now suppose that $y_c = 1$ for some class $c \notin \{i, j\}$. Then $p(y_i, y_j\,|\,\mathbf{y}_{\neg ij}, \mathbf{f}) = 1$ due to the one-hot constraint. Otherwise either $y_i = 1$ or $y_j = 1$. In this case, assume without loss of generality that $y_i = 1$ and $y_j = 0$ and thus

$$p(y_i, y_j\,|\,\mathbf{y}_{\neg ij}, \mathbf{f}) = \frac{e^{f_i}}{e^{f_i} + e^{f_j}} = \sigma(f_i - f_j). \tag{9}$$

The composite likelihood defined in this way with unit component weights is therefore

$$\mathcal{L}_{\text{OVE}}(\mathbf{f}\,|\,\mathbf{y}) = \prod_i \prod_{j \neq i} p(y_i, y_j | \mathbf{y}_{\neg ij}, \mathbf{f}) = \prod_i \prod_{j \neq i} \sigma(f_i - f_j)^{y_i}. \tag{10}$$

Alternatively, we may simply write $\mathcal{L}_{\text{OVE}}(\mathbf{f}\,|\,y = i) = \prod_{j \neq i} \sigma(f_i - f_j)$, which is identical to the OVE bound (6).

## 4.2 GP Classification with the OVE Likelihood

We now turn our attention to GP classification. Suppose we have access to examples $\mathbf{X} \in \mathbb{R}^{N \times D}$ with corresponding one-hot labels $\mathbf{Y} \in \{0,1\}^{N \times C}$, where $C$ is the number of classes. We consider the logits jointly as a single vector

$$\mathbf{f} \triangleq (f_1^1, \ldots, f_N^1, f_1^2, \ldots, f_N^2, \ldots, f_1^C, \ldots, f_N^C)^\top \tag{11}$$

and place an independent GP prior on the logits for each class: $\mathbf{f}^c(\mathbf{x}) \sim \mathcal{GP}(m(\mathbf{x}), k(\mathbf{x}, \mathbf{x}'))$. Therefore we have $p(\mathbf{f}|\mathbf{X}) = \mathcal{N}(\mathbf{f}|\boldsymbol{\mu}, \mathbf{K})$, where $\mu_i^c = m(\mathbf{x}_i)$ and $\mathbf{K}$ is block diagonal with $K_{ij}^c = k(\mathbf{x}_i, \mathbf{x}_j)$ for each block $\mathbf{K}^c$.

The Pólya-Gamma integral identity used to derive (2) does not have a multi-class analogue and thus a direct application of the augmentation scheme to the softmax likelihood is nontrivial. Instead, we propose to directly replace the softmax with the OVE-based composite likelihood function from (10) with unit weights. The posterior over $\mathbf{f}$ when using OVE as the likelihood function can be expressed as:

$$p(\mathbf{f}|\mathbf{X}, \mathbf{y}) \propto p(\mathbf{f}|\mathbf{X}) \prod_{i=1}^N \prod_{c' \neq y_i} \sigma(f_i^{y_i} - f_i^{c'}), \tag{12}$$

to which Pólya-Gamma augmentation can be applied as we show in the next section. Our motivation for using a composite likelihood therefore differs from the traditional motivation, which is to avoid the use of a likelihood function which is intractable to *evaluate*. Instead, we employ a composite likelihood because it makes posterior *inference* tractable when coupled with Pólya-Gamma augmentation.

Prior work on Bayesian inference with composite likelihoods has shown that the composite posterior is consistent under fairly general conditions for correctly specified models (Miller, 2019) but can produce overly concentrated posteriors (Pauli et al., 2011; Ribatet et al., 2012) since each component likelihood event is treated as independent when in reality there may be significant dependencies. Nevertheless, we show in Section 5 that in practice our method exhibits competitive accuracy and strong calibration relative to baseline few-shot learning algorithms. We leave further theoretical analysis of the OVE composite posterior and its properties for future work.

Compared to choices of likelihoods used by previous approaches, there are several reasons to prefer OVE. Relative to the Gaussian augmentation approach of Girolami & Rogers (2006), Pólya-Gamma augmentation has the benefit of fast mixing and the ability of a single value of $\boldsymbol{\omega}$ to capture much of the marginal distribution over function values[1]. The stick-breaking construction of Linderman et al. (2015) induces a dependence on the ordering of classes, which leads to undesirable asymmetry. Finally, the logistic-softmax likelihood of Galy-Fajou et al. (2020) requires three augmentations and careful learning of the mean function to avoid *a priori* underconfidence (see Section F.1 for more details).

### 4.3 POSTERIOR INFERENCE VIA GIBBS SAMPLING

We now describe how we perform tractable posterior inference in our model with Gibbs sampling. Define the matrix $\mathbf{A} \triangleq \text{OVE-MATRIX}(\mathbf{Y})$ to be a $CN \times CN$ sparse block matrix with $C$ row partitions and $C$ column partitions. Each block $\mathbf{A}_{cc'}$ is a diagonal $N \times N$ matrix defined as follows:

$$\mathbf{A}_{cc'} \triangleq \text{diag}(\mathbf{Y}_{\cdot c'}) - \mathbb{1}[c = c']\mathbf{I}_n, \tag{13}$$

where $\mathbf{Y}_{\cdot c'}$ denotes the $c'$th column of $\mathbf{Y}$. Now the binary logit vector $\boldsymbol{\psi} \triangleq \mathbf{Af} \in \mathbb{R}^{CN}$ will have entries equal to $f_i^{y_i} - f_i^c$ for each unique combination of $c$ and $i$, of which there are $CN$ in total. The OVE composite likelihood can now be written as $\mathcal{L}(\boldsymbol{\psi}|\mathbf{Y}) = 2^N \prod_{j=1}^{NC} \sigma(\psi_j)$, where the $2^N$ term arises from the $N$ cases in which $\psi_j = 0$ due to comparing the ground truth logit with itself.

Analogous to (2), the likelihood of $\boldsymbol{\psi}$ conditioned on $\boldsymbol{\omega}$ and $\mathbf{Y}$ is proportional to a diagonal Gaussian:

$$\mathcal{L}(\boldsymbol{\psi}|\mathbf{Y}, \boldsymbol{\omega}) \propto \prod_{j=1}^{NC} e^{-\omega_j \psi_j^2/2} e^{\kappa_j \psi_j} \propto \mathcal{N}(\boldsymbol{\Omega}^{-1}\boldsymbol{\kappa}|\boldsymbol{\psi}, \boldsymbol{\Omega}^{-1}), \tag{14}$$

where $\kappa_j = 1/2$ and $\boldsymbol{\Omega} = \text{diag}(\boldsymbol{\omega})$. By exploiting the fact that $\boldsymbol{\psi} = \mathbf{Af}$, we can express the likelihood in terms of $\mathbf{f}$ and write down the conditional composite posterior as follows:

$$p(\mathbf{f}|\mathbf{X}, \mathbf{Y}, \boldsymbol{\omega}) \propto \mathcal{N}(\boldsymbol{\Omega}^{-1}\boldsymbol{\kappa}|\mathbf{Af}, \boldsymbol{\Omega}^{-1})\mathcal{N}(\mathbf{f}|\boldsymbol{\mu}, \mathbf{K}) \propto \mathcal{N}(\mathbf{f}|\tilde{\boldsymbol{\Sigma}}(\mathbf{K}^{-1}\boldsymbol{\mu} + \mathbf{A}^\top\boldsymbol{\kappa}), \tilde{\boldsymbol{\Sigma}}), \tag{15}$$

where $\tilde{\boldsymbol{\Sigma}} = (\mathbf{K}^{-1} + \mathbf{A}^\top\boldsymbol{\Omega}\mathbf{A})^{-1}$, which is an expression remarkably similar to (3). Analogous to (4), the conditional distribution over $\boldsymbol{\omega}$ given $\mathbf{f}$ and the data becomes $p(\boldsymbol{\omega}|\mathbf{y}, \mathbf{f}) = \text{PG}(\boldsymbol{\omega}|\mathbf{1}, \mathbf{Af})$.

The primary computational bottleneck of posterior inference lies in sampling $\mathbf{f}$ from (15). Since $\tilde{\boldsymbol{\Sigma}}$ is a $CN \times CN$ matrix, a naive implementation has complexity $\mathcal{O}(C^3N^3)$. By utilizing the matrix inversion lemma and Gaussian sampling techniques summarized in (Doucet, 2010), this can be brought down to $\mathcal{O}(CN^3)$. Details may be found in Section B.

### 4.4 LEARNING COVARIANCE HYPERPARAMETERS FOR FEW-SHOT CLASSIFICATION

We now describe how we apply OVE Pólya-Gamma augmented GPs to few-shot classification. We assume the standard episodic few-shot setup in which one observes a labeled support set $\mathcal{S} = (\mathbf{X}, \mathbf{Y})$. Predictions must then be made for a query example $(\mathbf{x}_*, \mathbf{y}_*)$. We consider a zero-mean GP prior over the class logits $\mathbf{f}^c(\mathbf{x}) \sim \mathcal{GP}(\mathbf{0}, k_{\boldsymbol{\theta}}(\mathbf{x}, \mathbf{x}'))$, where $\boldsymbol{\theta}$ are learnable parameters of our covariance function. These could include traditional hyperparameters such as lengthscales or the weights of a deep neural network as in deep kernel learning (Wilson et al., 2016).

We consider two objectives for learning hyperparameters of the covariance function: the marginal likelihood (ML) and the predictive likelihood (PL). Marginal likelihood measures the likelihood of the hyperparameters given the observed data and is intuitively appealing from a Bayesian perspective. On the other hand, many standard FSC methods optimize for predictive likelihood on the query set (Vinyals et al., 2016; Finn et al., 2017; Snell et al., 2017). Both objectives marginalize over latent functions, thereby making full use of our Bayesian formulation.

The details of these objectives and how we compute gradients can be found in Section C. Our learning algorithm for both marginal and predictive likelihood may be found in Section D. Details of computing the posterior predictive distribution $p(\mathbf{y}_*|\mathbf{x}_*, \mathbf{X}, \mathbf{Y}, \boldsymbol{\omega})$ may be found in Section E. Finally, details of our chosen "cosine" kernel may be found in Section H.

---

[1]See in particular Appendix C of (Linderman et al., 2015) for a detailed explanation of this phenomenon.

## 5 EXPERIMENTS

In this section, we present our results on few-shot classification both in terms of accuracy and uncertainty quantification. Additional results comparing the one-vs-each composite likelihood to the softmax, logistic softmax, and Gaussian likelihoods may be found in Section F.

One of our aims is to compare methods based on uncertainty quantification. We therefore developed new benchmark evaluations and tasks: few-shot calibration, robustness, and out-of-episode detection. In order to empirically compare methods, we could not simply borrow the accuracy results from other papers, but instead needed to train each of these baselines ourselves. For all baselines except Bayesian MAML, ABML, and Logistic Softmax GP, we ran the code from (Patacchiola et al., 2020) and verified that the accuracies matched closely to their reported results. We have made PyTorch code for our experiments publicly available[2].

### 5.1 FEW-SHOT CLASSIFICATION

For our few-shot classification experiments, we follow the training and evaluation protocol of Patacchiola et al. (2020). We train both 1-shot and 5-shot versions of our model in four different settings: Caltech-UCSD Birds (CUB) (Wah et al., 2011), mini-Imagenet with the split proposed by Ravi & Larochelle (2017), as well as two cross-domain transfer tasks. The first transfer task entails training on mini-ImageNet and testing on CUB, and the second measures transfer from Omniglot (Lake et al., 2011) to EMNIST (Cohen et al., 2017). Experimental details and an overview of the baselines we used can be found in Section G. Classification results are shown in Table 1 and 2. We find that our proposed Pólya-Gamma OVE GPs yield strong classification results, outperforming the baselines in five of the eight scenarios.

Table 1: Average accuracy and standard deviation (percentage) on 5-way FSC. Baseline results (through DKT) are from Patacchiola et al. (2020). Evaluation is performed on 3,000 randomly generated test episodes. Standard deviation for the remaining methods are computed by averaging over 5 batches of 600 episodes with different random seeds. The best results are highlighted in bold.

| | CUB | | mini-ImageNet | |
|---|---|---|---|---|
| **Method** | **1-shot** | **5-shot** | **1-shot** | **5-shot** |
| Feature Transfer | $46.19 \pm 0.64$ | $68.40 \pm 0.79$ | $39.51 \pm 0.23$ | $60.51 \pm 0.55$ |
| Baseline++ | $61.75 \pm 0.95$ | $78.51 \pm 0.59$ | $47.15 \pm 0.49$ | $66.18 \pm 0.18$ |
| MatchingNet | $60.19 \pm 1.02$ | $75.11 \pm 0.35$ | $48.25 \pm 0.65$ | $62.71 \pm 0.44$ |
| ProtoNet | $52.52 \pm 1.90$ | $75.93 \pm 0.46$ | $44.19 \pm 1.30$ | $64.07 \pm 0.65$ |
| RelationNet | $62.52 \pm 0.34$ | $78.22 \pm 0.07$ | $48.76 \pm 0.17$ | $64.20 \pm 0.28$ |
| MAML | $56.11 \pm 0.69$ | $74.84 \pm 0.62$ | $45.39 \pm 0.49$ | $61.58 \pm 0.53$ |
| DKT + Cosine | $63.37 \pm 0.19$ | $77.73 \pm 0.26$ | $48.64 \pm 0.45$ | $62.85 \pm 0.37$ |
| Bayesian MAML | $55.93 \pm 0.71$ | $72.87 \pm 0.26$ | $44.46 \pm 0.30$ | $62.60 \pm 0.25$ |
| Bayesian MAML (Chaser) | $53.93 \pm 0.72$ | $71.16 \pm 0.32$ | $43.74 \pm 0.46$ | $59.23 \pm 0.34$ |
| ABML | $48.80 \pm 0.40$ | $70.91 \pm 0.32$ | $40.88 \pm 0.25$ | $58.19 \pm 0.17$ |
| Logistic Softmax GP + Cosine (ML) | $60.23 \pm 0.54$ | $74.58 \pm 0.25$ | $46.75 \pm 0.20$ | $59.93 \pm 0.31$ |
| Logistic Softmax GP + Cosine (PL) | $60.07 \pm 0.29$ | $78.14 \pm 0.07$ | $47.05 \pm 0.20$ | $66.01 \pm 0.25$ |
| OVE PG GP + Cosine (ML) [ours] | $\mathbf{63.98 \pm 0.43}$ | $77.44 \pm 0.18$ | $\mathbf{50.02 \pm 0.35}$ | $64.58 \pm 0.31$ |
| OVE PG GP + Cosine (PL) [ours] | $60.11 \pm 0.26$ | $\mathbf{79.07 \pm 0.05}$ | $48.00 \pm 0.24$ | $\mathbf{67.14 \pm 0.23}$ |

### 5.2 UNCERTAINTY QUANTIFICATION THROUGH CALIBRATION

We next turn to uncertainty quantification, an important concern for few-shot classifiers. When used in safety-critical applications such as medical diagnosis, it is important for a machine learning system to defer when there is not enough evidence to make a decision. Even in non-critical applications, precise uncertainty quantification helps practitioners in the few-shot setting determine when a class has an adequate amount of labeled data or when more labels are required, and can facilitate active learning.

---

[2]https://github.com/jakesnell/ove-polya-gamma-gp

Table 2: Average accuracy and standard deviation (percentage) on 5-way cross-domain FSC, with the same experimental setup as in Table 1. Baseline results (through DKT) are from (Patacchiola et al., 2020).

| Method | Omniglot→EMNIST | | mini-ImageNet→CUB | |
| --- | --- | --- | --- | --- |
| | 1-shot | 5-shot | 1-shot | 5-shot |
| **Feature Transfer** | $64.22 \pm 1.24$ | $86.10 \pm 0.84$ | $32.77 \pm 0.35$ | $50.34 \pm 0.27$ |
| **Baseline++** | $56.84 \pm 0.91$ | $80.01 \pm 0.92$ | $39.19 \pm 0.12$ | $\mathbf{57.31 \pm 0.11}$ |
| **MatchingNet** | $75.01 \pm 2.09$ | $87.41 \pm 1.79$ | $36.98 \pm 0.06$ | $50.72 \pm 0.36$ |
| **ProtoNet** | $72.04 \pm 0.82$ | $87.22 \pm 1.01$ | $33.27 \pm 1.09$ | $52.16 \pm 0.17$ |
| **RelationNet** | $75.62 \pm 1.00$ | $87.84 \pm 0.27$ | $37.13 \pm 0.20$ | $51.76 \pm 1.48$ |
| **MAML** | $72.68 \pm 1.85$ | $83.54 \pm 1.79$ | $34.01 \pm 1.25$ | $48.83 \pm 0.62$ |
| **DKT + Cosine** | $73.06 \pm 2.36$ | $\mathbf{88.10 \pm 0.78}$ | $\mathbf{40.22 \pm 0.54}$ | $55.65 \pm 0.05$ |
| **Bayesian MAML** | $63.94 \pm 0.47$ | $65.26 \pm 0.30$ | $33.52 \pm 0.36$ | $51.35 \pm 0.16$ |
| **Bayesian MAML (Chaser)** | $55.04 \pm 0.34$ | $54.19 \pm 0.32$ | $36.22 \pm 0.50$ | $51.53 \pm 0.43$ |
| **ABML** | $73.89 \pm 0.24$ | $87.28 \pm 0.40$ | $31.51 \pm 0.32$ | $47.80 \pm 0.51$ |
| **Logistic Softmax GP + Cosine (ML)** | $62.91 \pm 0.49$ | $83.80 \pm 0.13$ | $36.41 \pm 0.18$ | $50.33 \pm 0.13$ |
| **Logistic Softmax GP + Cosine (PL)** | $70.70 \pm 0.36$ | $86.59 \pm 0.15$ | $36.73 \pm 0.26$ | $56.70 \pm 0.31$ |
| **OVE PG GP + Cosine (ML)** [ours] | $68.43 \pm 0.67$ | $86.22 \pm 0.20$ | $39.66 \pm 0.18$ | $55.71 \pm 0.31$ |
| **OVE PG GP + Cosine (PL)** [ours] | $\mathbf{77.00 \pm 0.50}$ | $87.52 \pm 0.19$ | $37.49 \pm 0.11$ | $57.23 \pm 0.31$ |

We chose several commonly used metrics for calibration. Expected calibration error (ECE) (Guo et al., 2017) measures the expected binned difference between confidence and accuracy. Maximum calibration error (MCE) is similar to ECE but measures maximum difference instead of expected difference. Brier score (BRI) (Brier, 1950) is a proper scoring rule computed as the squared error between the output probabilities and the one-hot label. For a recent perspective on metrics for uncertainty evaluation, please refer to Ovadia et al. (2019). The results for representative approaches on 5-shot, 5-way CUB can be found in Figure 1. Our OVE PG GPs are the best calibrated overall across the metrics.

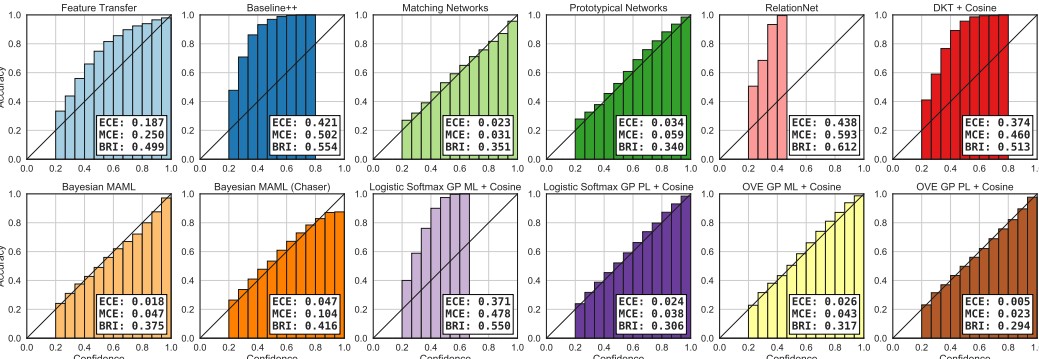

Figure 1: Reliability diagrams, expected calibration error (ECE), maximum calibration error (MCE), and Brier Score (BRI) for 5-shot 5-way tasks on CUB (additional calibration results can be found in Appendix I). Metrics are computed on 3,000 random tasks from the test set. The last two plots are our proposed method.

## 5.3 ROBUSTNESS TO INPUT NOISE

Input examples for novel classes in FSC may have been collected under conditions that do not match those observed at training time. For example, labeled support images in a medical diagnosis application may come from a different hospital than the training set. To mimic a simplified version of this scenario, we investigate robustness to input noise. We used the Imagecorruptions package (Michaelis et al., 2019) to apply Gaussian noise, impulse noise, and defocus blur to both the support set and query sets of episodes at test time and evaluated both accuracy and calibration. We used corruption severity of 5 (severe) and evaluated across 1,000 randomly generated tasks on the three

datasets involving natural images. The robustness results for Gaussian noise are shown in Figure 2. Full quantitative results tables for each noise type may be found in Section J. We find that in general Bayesian approaches tend to be robust due to their ability to marginalize over hypotheses consistent with the support labels. Our approach is one of the top performing methods across all settings.

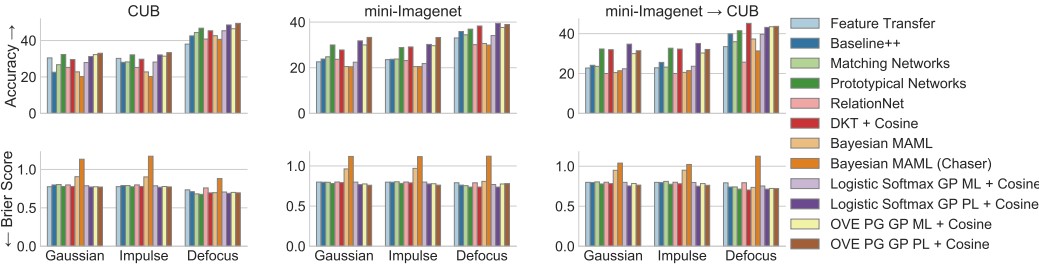

Figure 2: Accuracy (↑) and Brier Score (↓) when corrupting both support and query with Gaussian noise on 5-way 5-shot tasks. Quantitative results may be found in Appendix J.

## 5.4 OUT-OF-EPISODE DETECTION

Finally, we measure performance on out-of-episode detection, another application in which uncertainty quantification is important. In this experiment, we used 5-way, 5-shot support sets at test time but incorporated out-of-episode examples into the query set. Each episode had 150 query examples: 15 from each of 5 randomly chosen in-episode classes and 15 from each of 5 randomly chosen out-of-episode classes. We then computed the AUROC of binary outlier detection using the negative of the maximum logit as the score. Intuitively, if none of the support classes assign a high logit to the example, it can be classified as an outlier. The results are shown in Figure 3. Our approach generally performs the best across the datasets.

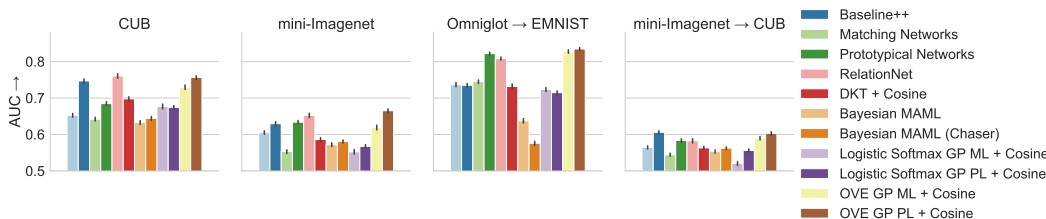

Figure 3: Average AUROC (↑) for out-of-episode detection. The AUC is computed separately for each episode and averaged across 1,000 episodes. Bars indicate a 95% bootstrapped confidence interval.

## 6 CONCLUSION

In this work, we have proposed a Bayesian few-shot classification approach based on Gaussian processes. Our method replaces the ordinary softmax likelihood with a one-vs-each pairwise composite likelihood and applies Pólya-Gamma augmentation to perform inference. This allows us to model class logits directly as function values and efficiently marginalize over uncertainty in each few-shot episode. Modeling functions directly enables our approach to avoid the dependence on model size that posterior inference in weight-space based models inherently have. Our approach compares favorably to baseline FSC methods under a variety of dataset and shot configurations, including dataset transfer. We also demonstrate strong uncertainty quantification, robustness to input noise, and out-of-episode detection. We believe that Bayesian modeling is a powerful tool for handling uncertainty and hope that our work will lead to broader adoption of efficient Bayesian inference in the few-shot scenario.

ACKNOWLEDGMENTS

We would like to thank Ryan Adams, Ethan Fetaya, Mike Mozer, Eleni Triantafillou, Kuan-Chieh Wang, and Max Welling for helpful discussions. JS also thanks SK T-Brain for supporting him on an internship that led to precursors of some ideas in this paper. Resources used in preparing this research were provided, in part, by the Province of Ontario, the Government of Canada through CIFAR, and companies sponsoring the Vector Institute (`https://www.vectorinstitute.ai/partners`). This project is supported by NSERC and the Intelligence Advanced Research Projects Activity (IARPA) via Department of Interior/Interior Business Center (DoI/IBC) contract number D16PC00003. The U.S. Government is authorized to reproduce and distribute reprints for Governmental purposes notwithstanding any copyright annotation thereon. Disclaimer: The views and conclusions contained herein are those of the authors and should not be interpreted as necessarily representing the official policies or endorsements, either expressed or implied, of IARPA, DoI/IBC, or the U.S. Government.

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

## A    Derivation of Pólya-Gamma Augmented Logistic Likelihood

In this section, we show the derivation for the augmented logistic likelihood presented in Section 3.1. First, recall the logistic likelihood:

$$p(\mathbf{y}|\boldsymbol{\psi}) = \prod_{i=1}^{N} \sigma(\psi_i)^{y_i}(1 - \sigma(\psi_i))^{1-y_i} = \prod_{i=1}^{N} \frac{(e^{\psi_i})^{y_i}}{1 + e^{\psi_i}}, \tag{16}$$

where $\sigma(\cdot)$ is the logistic sigmoid function. We have a Gaussian prior $p(\boldsymbol{\psi}) = \mathcal{N}(\boldsymbol{\psi}|\boldsymbol{\mu}, \boldsymbol{\Sigma})$ and introduce Pólya-Gamma auxiliary random variables $\boldsymbol{\omega}$ to the likelihood such that the original model is recovered when $\boldsymbol{\omega}$ is marginalized out: $p(\mathbf{y}|\boldsymbol{\psi}) = \int p(\boldsymbol{\omega})p(\mathbf{y}|\boldsymbol{\psi}, \boldsymbol{\omega}) \, d\boldsymbol{\omega}$.

The Pólya-Gamma distribution $\omega \sim \mathrm{PG}(b, c)$ can be written as an infinite convolution of Gamma distributions:

$$\omega \overset{D}{=} \frac{1}{2\pi^2} \sum_{k=1}^{\infty} \frac{\mathrm{Ga}(b, 1)}{(k - 1/2)^2 + c^2/(4\pi^2)}. \tag{17}$$

The following integral identity holds for $b > 0$:

$$\frac{(e^{\psi})^a}{(1 + e^{\psi})^b} = 2^{-b}e^{\kappa\psi} \int_0^{\infty} e^{-\omega\psi^2/2}p(\omega) \, d\omega, \tag{18}$$

where $\kappa = a - b/2$ and $\omega \sim \mathrm{PG}(b, 0)$. Specifically, when $a = y$ and $b = 1$, we recover an individual term of the logistic likelihood (16):

$$p(y|\psi) = \frac{(e^{\psi})^y}{1 + e^{\psi}} = \frac{1}{2}e^{\kappa\psi} \int_0^{\infty} e^{-\omega\psi^2/2}p(\omega) \, d\omega, \tag{19}$$

where $\kappa = y - 1/2$ and $\omega \sim PG(1, 0)$. Conditioned on $\boldsymbol{\omega}$, the batch likelihood is proportional to a diagonal Gaussian:

$$p(\mathbf{y}|\boldsymbol{\psi}, \boldsymbol{\omega}) \propto \prod_{i=1}^{N} e^{-\omega_i\psi_i^2/2}e^{\kappa_i\psi_i} \propto \mathcal{N}(\boldsymbol{\Omega}^{-1}\boldsymbol{\kappa} \,|\, \boldsymbol{\psi}, \boldsymbol{\Omega}^{-1}), \tag{20}$$

where $\kappa_i = y_i - 1/2$ and $\boldsymbol{\Omega} = \mathrm{diag}(\boldsymbol{\omega})$. The conditional distribution over $\boldsymbol{\psi}$ given $\mathbf{y}$ and $\boldsymbol{\omega}$ is now tractable:

$$p(\boldsymbol{\psi}|\mathbf{y}, \boldsymbol{\omega}) \propto p(\mathbf{y}|\boldsymbol{\psi}, \boldsymbol{\omega})p(\boldsymbol{\psi}) \propto \mathcal{N}(\boldsymbol{\psi}|\tilde{\boldsymbol{\Sigma}}(\boldsymbol{\Sigma}^{-1}\boldsymbol{\mu} + \boldsymbol{\kappa}), \tilde{\boldsymbol{\Sigma}}), \tag{21}$$

where $\tilde{\boldsymbol{\Sigma}} = (\boldsymbol{\Sigma}^{-1} + \boldsymbol{\Omega})^{-1}$.

## B    Efficient Gibbs Sampling

The Gibbs conditional distribution over $\mathbf{f}$ is given by:

$$p(\mathbf{f}|\mathbf{X}, \mathbf{y}, \boldsymbol{\omega}) = \mathcal{N}(\mathbf{f}|\tilde{\boldsymbol{\Sigma}}(\mathbf{K}^{-1}\boldsymbol{\mu} + \mathbf{A}^{\top}\boldsymbol{\kappa}), \tilde{\boldsymbol{\Sigma}}), \tag{22}$$

where $\tilde{\boldsymbol{\Sigma}} = (\mathbf{K}^{-1} + \mathbf{A}^{\top}\boldsymbol{\Omega}\mathbf{A})^{-1}$. Naively sampling from this distribution requires $\mathcal{O}(C^3N^3)$ computation since $\tilde{\boldsymbol{\Sigma}}$ is a $CN \times CN$ matrix. Here we describe a method for sampling from this distribution that requires $\mathcal{O}(CN^3)$ computation instead.

First, we note that (22) can be interpreted as the conditional distribution $p(\mathbf{f}|\mathbf{z} = \boldsymbol{\Omega}^{-1}\boldsymbol{\kappa})$ resulting from the following marginal distribution $p(\mathbf{f})$ and conditional $p(\mathbf{z}|\mathbf{f})$:

$$p(\mathbf{f}) = \mathcal{N}(\mathbf{f}|\boldsymbol{\mu}, \mathbf{K}) \tag{23}$$

$$p(\mathbf{z}|\mathbf{f}) = \mathcal{N}(\mathbf{z}|\mathbf{A}\mathbf{f}, \boldsymbol{\Omega}^{-1}), \tag{24}$$

where we have made implicit the dependence on $\mathbf{X}, \mathbf{Y}$, and $\boldsymbol{\omega}$ for brevity of notation. Equivalently, the distribution over $\mathbf{f}$ and $\mathbf{z}$ can be represented by the partitioned Gaussian

$$\begin{bmatrix} \mathbf{f} \\ \mathbf{z} \end{bmatrix} \sim \mathcal{N}\left( \begin{bmatrix} \boldsymbol{\mu} \\ \mathbf{A}\boldsymbol{\mu} \end{bmatrix}, \begin{bmatrix} \mathbf{K} & \mathbf{K}\mathbf{A}^{\top} \\ \mathbf{A}\mathbf{K} & \mathbf{A}\mathbf{K}\mathbf{A}^{\top} + \boldsymbol{\Omega}^{-1} \end{bmatrix} \right). \tag{25}$$

The conditional distribution $p(\mathbf{f}|\mathbf{z})$ is given as:

$$p(\mathbf{f}|\mathbf{z}) = \mathcal{N}(\mathbf{f}|\tilde{\boldsymbol{\Sigma}}(\mathbf{K}^{-1}\boldsymbol{\mu} + \mathbf{A}^{\top}\boldsymbol{\Omega}\mathbf{z}), \tilde{\boldsymbol{\Sigma}}), \tag{26}$$

where $\tilde{\boldsymbol{\Sigma}} = (\mathbf{K}^{-1} + \mathbf{A}^{\top}\boldsymbol{\Omega}\mathbf{A})^{-1}$. Note that $p(\mathbf{f}|\mathbf{z} = \boldsymbol{\Omega}^{-1}\boldsymbol{\kappa})$ recovers our desired Gibbs conditional distribution from (22).

An efficient approach to conditional Gaussian sampling is due to Hoffman & Ribak (1991) and described in greater clarity by Doucet (2010). The procedure is as follows:

1. Sample $\mathbf{f}_0 \sim p(\mathbf{f})$ and $\mathbf{z}_0 \sim p(\mathbf{z}|\mathbf{f})$.
2. Return $\bar{\mathbf{f}} = \mathbf{f}_0 + \mathbf{K}\mathbf{A}^{\top}(\mathbf{A}\mathbf{K}\mathbf{A}^{\top} + \boldsymbol{\Omega}^{-1})^{-1}(\boldsymbol{\Omega}^{-1}\boldsymbol{\kappa} - \mathbf{z}_0)$ as the sample from $p(\mathbf{f}|\mathbf{z})$.

$\mathbf{K}$ is block diagonal and thus sampling from $p(\mathbf{f})$ requires $\mathcal{O}(CN^3)$ time. $\mathbf{A}\mathbf{f}$ can be computed in $\mathcal{O}(CN)$ time, since each entry is the difference between $f_i^{y_i}$ and $f_i^c$ for some $i$ and $c$. Overall, step 1 requires $\mathcal{O}(CN^3)$ time.

We now show how to compute $\bar{\mathbf{f}}$ from step 2 in $\mathcal{O}(CN^3)$ time. We first expand $(\mathbf{A}\mathbf{K}\mathbf{A}^{\top} + \boldsymbol{\Omega}^{-1})^{-1}$:

$$(\mathbf{A}\mathbf{K}\mathbf{A}^{\top} + \boldsymbol{\Omega}^{-1})^{-1} = \boldsymbol{\Omega} - \boldsymbol{\Omega}\mathbf{A}(\mathbf{K}^{-1} + \mathbf{A}^{\top}\boldsymbol{\Omega}\mathbf{A})^{-1}\mathbf{A}^{\top}\boldsymbol{\Omega} \tag{27}$$

We substitute into the expression for $\bar{\mathbf{f}}$:

$$\bar{\mathbf{f}} = \mathbf{f}_0 + \mathbf{K}\mathbf{A}^{\top}(\boldsymbol{\Omega} - \boldsymbol{\Omega}\mathbf{A}(\mathbf{K}^{-1} + \mathbf{A}^{\top}\boldsymbol{\Omega}\mathbf{A})^{-1}\mathbf{A}^{\top}\boldsymbol{\Omega})(\boldsymbol{\Omega}^{-1}\boldsymbol{\kappa} - \mathbf{z}_0) \tag{28}$$

$$= \mathbf{f}_0 + \mathbf{K}\mathbf{A}^{\top}\boldsymbol{\Omega}(\boldsymbol{\Omega}^{-1}\boldsymbol{\kappa} - \mathbf{z}_0) - \mathbf{K}\mathbf{A}^{\top}\boldsymbol{\Omega}\mathbf{A}(\mathbf{K}^{-1} + \mathbf{A}^{\top}\boldsymbol{\Omega}\mathbf{A})^{-1}\mathbf{A}^{\top}\boldsymbol{\Omega}(\boldsymbol{\Omega}^{-1}\boldsymbol{\kappa} - \mathbf{z}_0) \tag{29}$$

$$= \mathbf{f}_0 + \mathbf{K}\mathbf{v} - \mathbf{K}\mathbf{A}^{\top}\boldsymbol{\Omega}\mathbf{A}(\mathbf{K}^{-1} + \mathbf{A}^{\top}\boldsymbol{\Omega}\mathbf{A})^{-1}\mathbf{v}, \tag{30}$$

where we have defined $\mathbf{v} \triangleq \mathbf{A}^{\top}\boldsymbol{\Omega}(\boldsymbol{\Omega}^{-1}\boldsymbol{\kappa} - \mathbf{z}_0)$.

Now let $\mathbf{d} \triangleq (d_1^1, \ldots, d_N^1, d_1^2, \ldots, d_N^2, \ldots, d_1^C, \ldots, d_N^C)^{\top}$, where $d_i^c = Y_{ic}\sum_{c'}\omega_i^{c'}$. Define $\mathbf{Y}^{\dagger}$ to be the $CN \times N$ matrix produced by vertically stacking $\mathrm{diag}(Y_{\cdot c})$, and let $\mathbf{W}^{\dagger}$ be the $CN \times N$ matrix produced by vertically stacking $\mathrm{diag}((\omega_1^c, \ldots, \omega_N^c)^{\top})$. $\mathbf{A}^{\top}\boldsymbol{\Omega}\mathbf{A}$ may then be written as follows:

$$\mathbf{A}^{\top}\boldsymbol{\Omega}\mathbf{A} = \mathbf{D} - \mathbf{S}\mathbf{P}\mathbf{S}^{\top}, \text{ where} \tag{31}$$

$$\mathbf{D} = \boldsymbol{\Omega} + \mathrm{diag}(\mathbf{d}), \tag{32}$$

$$\mathbf{S} = \begin{bmatrix} \mathbf{Y}^{\dagger} & \mathbf{W}^{\dagger} \end{bmatrix}, \tag{33}$$

$$\mathbf{P} = \begin{bmatrix} \mathbf{0}_N & \mathbf{I}_N \\ \mathbf{I}_N & \mathbf{0}_N \end{bmatrix}. \tag{34}$$

Substituting (31) into (30):

$$\bar{\mathbf{f}} = \mathbf{f}_0 + \mathbf{K}\mathbf{v} - \mathbf{K}\mathbf{A}^{\top}\boldsymbol{\Omega}\mathbf{A}(\mathbf{K}^{-1} + \mathbf{D} - \mathbf{S}\mathbf{P}\mathbf{S}^{\top})^{-1}\mathbf{v}. \tag{35}$$

Now we expand $(\mathbf{K}^{-1} + \mathbf{D} - \mathbf{S}\mathbf{P}\mathbf{S}^{\top})^{-1}$ :

$$(\mathbf{K}^{-1} + \mathbf{D} - \mathbf{S}\mathbf{P}\mathbf{S}^{\top})^{-1} = \mathbf{E} - \mathbf{E}\mathbf{S}(\mathbf{S}^{\top}\mathbf{E}\mathbf{S} - \mathbf{P}^{-1})^{-1}\mathbf{S}^{\top}\mathbf{E}, \tag{36}$$

where $\mathbf{E} = (\mathbf{K}^{-1} + \mathbf{D})^{-1} = \mathbf{K}(\mathbf{K} + \mathbf{D}^{-1})^{-1}\mathbf{D}^{-1}$ is a block-diagonal matrix that can be computed in $\mathcal{O}(CN^3)$ time, since $\mathbf{D}$ is diagonal and $\mathbf{K}$ is block diagonal. Now, substituting (36) back into (35),

$$\bar{\mathbf{f}} = \mathbf{f}_0 + \mathbf{K}\mathbf{v} - \mathbf{K}\mathbf{A}^{\top}\boldsymbol{\Omega}\mathbf{A}\mathbf{E}\mathbf{v} + \mathbf{K}\mathbf{A}^{\top}\boldsymbol{\Omega}\mathbf{A}\mathbf{E}\mathbf{S}(\mathbf{S}^{\top}\mathbf{E}\mathbf{S} - \mathbf{P}^{-1})^{-1}\mathbf{S}^{\top}\mathbf{E}\mathbf{v}. \tag{37}$$

Note that $(\mathbf{S}^{\top}\mathbf{E}\mathbf{S} - \mathbf{P}^{-1})^{-1}$ is a $2N \times 2N$ matrix and thus can be inverted in $\mathcal{O}(N^3)$ time. The overall complexity is therefore $\mathcal{O}(CN^3)$.

## C MARGINAL LIKELIHOOD AND PREDICTIVE LIKELIHOOD OBJECTIVES

**Marginal Likelihood (ML).** The log marginal likelihood can be written as follows:

$$L_{\mathrm{ML}}(\boldsymbol{\theta}; \mathbf{X}, \mathbf{Y}) \triangleq \log p_{\boldsymbol{\theta}}(\mathbf{Y}|\mathbf{X}) = \log \int p(\boldsymbol{\omega})p_{\boldsymbol{\theta}}(\mathbf{Y}|\boldsymbol{\omega}, \mathbf{X}) \, d\boldsymbol{\omega}$$

$$= \log \int p(\boldsymbol{\omega}) \int \mathcal{L}(\mathbf{f}|\mathbf{Y}, \boldsymbol{\omega})p_{\boldsymbol{\theta}}(\mathbf{f}|\mathbf{X}) \, d\mathbf{f} \, d\boldsymbol{\omega} \tag{38}$$

The gradient of the log marginal likelihood can be estimated by posterior samples $\boldsymbol{\omega} \sim p_{\boldsymbol{\theta}}(\boldsymbol{\omega}|\mathbf{X}, \mathbf{Y})$. In practice, we use a stochastic training objective based on samples of $\boldsymbol{\omega}$ from Gibbs chains. We use Fisher's identity (Douc et al., 2014) to derive the following gradient estimator:

$$\nabla_{\boldsymbol{\theta}} L_{\text{ML}} = \int p_{\boldsymbol{\theta}}(\boldsymbol{\omega}|\mathbf{X}, \mathbf{Y}) \nabla_{\boldsymbol{\theta}} \log p_{\boldsymbol{\theta}}(\mathbf{Y}|\boldsymbol{\omega}, \mathbf{X}) \, d\boldsymbol{\omega} \approx \frac{1}{M} \sum_{m=1}^{M} \nabla_{\boldsymbol{\theta}} \log p_{\boldsymbol{\theta}}(\mathbf{Y}|\mathbf{X}, \boldsymbol{\omega}^{(m)}), \quad (39)$$

where $\boldsymbol{\omega}^{(1)}, \ldots, \boldsymbol{\omega}^{(M)}$ are samples from the posterior Gibbs chain. As suggested by Patacchiola et al. (2020), who applied GPs to FSC via least-squares classification, we merge the support and query sets during learning to take full advantage of the available data within each episode.

**Predictive Likelihood (PL).** The log predictive likelihood for a query example $\mathbf{x}_*$ is:

$$L_{\text{PL}}(\boldsymbol{\theta}; \mathbf{X}, \mathbf{Y}, \mathbf{x}_*, \mathbf{y}_*) \triangleq \log p_{\boldsymbol{\theta}}(\mathbf{y}_*|\mathbf{x}_*, \mathbf{X}, \mathbf{Y}) = \log \int p(\boldsymbol{\omega}) p_{\boldsymbol{\theta}}(\mathbf{y}_*|\mathbf{x}_*, \mathbf{X}, \mathbf{Y}, \boldsymbol{\omega}) \, d\boldsymbol{\omega}. \quad (40)$$

We use an approximate gradient estimator again based on posterior samples of $\boldsymbol{\omega}$:

$$\nabla_{\boldsymbol{\theta}} L_{\text{PL}} \approx \int p_{\boldsymbol{\theta}}(\boldsymbol{\omega}|\mathbf{X}, \mathbf{Y}) \nabla_{\boldsymbol{\theta}} \log p_{\boldsymbol{\theta}}(\mathbf{y}_*|\mathbf{x}_*, \mathbf{X}, \mathbf{Y}) \, d\boldsymbol{\omega} \approx \frac{1}{M} \sum_{m=1}^{M} \nabla_{\boldsymbol{\theta}} \log p_{\boldsymbol{\theta}}(\mathbf{y}_*|\mathbf{x}_*, \mathbf{X}, \mathbf{Y}, \boldsymbol{\omega}^{(m)}).$$
$$(41)$$

We note that this is not an unbiased estimator of the gradient, but find it works well in practice.

## D  LEARNING ALGORITHM

Our learning algorithm for both marginal and predictive likelihood is summarized in Algorithm 1.

---

**Algorithm 1** One-vs-Each Pólya-Gamma GP Learning

---

**Input:** Objective $L \in \{L_{\text{ML}}, L_{\text{PL}}\}$, Task distribution $\mathcal{T}$, number of parallel Gibbs chains $M$, number of steps $T$, learning rate $\eta$.
Initialize hyperparameters $\boldsymbol{\theta}$ randomly.
**repeat**
  Sample $\mathcal{S} = (\mathbf{X}, \mathbf{Y}), \mathcal{Q} = (\mathbf{X}_*, \mathbf{Y}_*) \sim \mathcal{T}$
  **if** $L = L_{\text{ML}}$ **then**
    $\mathbf{X} \leftarrow \mathbf{X} \cup \mathbf{X}_*, \mathbf{Y} \leftarrow \mathbf{Y} \cup \mathbf{Y}_*$
  **end if**
  $\mathbf{A} \leftarrow \text{OVE-MATRIX}(\mathbf{Y})$
  **for** $m = 1$ **to** $M$ **do**
    $\boldsymbol{\omega}_0^{(m)} \sim PG(1, 0), \mathbf{f}_0^{(m)} \sim p_{\boldsymbol{\theta}}(\mathbf{f}|\mathbf{X})$
    **for** $t = 1$ **to** $T$ **do**
      $\boldsymbol{\psi}_t^{(m)} \leftarrow \mathbf{A} \mathbf{f}_{t-1}^{(m)}$
      $\boldsymbol{\omega}_t^{(m)} \sim PG(1, \boldsymbol{\psi}_t^{(m)})$
      $\mathbf{f}_t^{(m)} \sim p_{\boldsymbol{\theta}}(\mathbf{f}|\mathbf{X}, \mathbf{Y}, \boldsymbol{\omega}_t^{(m)})$
    **end for**
  **end for**
  **if** $L = L_{\text{ML}}$ **then**
    $\boldsymbol{\theta} \leftarrow \boldsymbol{\theta} + \frac{\eta}{M} \sum_{m=1}^{M} \nabla_{\boldsymbol{\theta}} \log p_{\boldsymbol{\theta}}(\mathbf{Y}|\mathbf{X}, \boldsymbol{\omega}_T^{(m)})$
  **else**
    $\boldsymbol{\theta} \leftarrow \boldsymbol{\theta} + \frac{\eta}{M} \sum_{m=1}^{M} \sum_j \nabla_{\boldsymbol{\theta}} \log p_{\boldsymbol{\theta}}(\mathbf{y}_{*j}|\mathbf{x}_{*j}, \mathcal{S}, \boldsymbol{\omega}_T^{(m)})$
  **end if**
**until** convergence

---

## E  POSTERIOR PREDICTIVE DISTRIBUTION

The posterior predictive distribution for a query example $\mathbf{x}_*$ conditioned on $\boldsymbol{\omega}$ is:

$$p(\mathbf{y}_*|\mathbf{x}_*, \mathbf{X}, \mathbf{Y}, \boldsymbol{\omega}) = \int p(\mathbf{y}_*|\mathbf{f}_*) p(\mathbf{f}_*|\mathbf{x}_*, \mathbf{X}, \mathbf{Y}, \boldsymbol{\omega}) \, d\mathbf{f}_*, \quad (42)$$

where $\mathbf{f}_*$ are the query example's logits. The predictive distribution over $\mathbf{f}_*$ can be obtained by noting that $\psi$ and the query logits are jointly Gaussian:

$$\begin{bmatrix} \psi \\ \mathbf{f}_* \end{bmatrix} \sim \mathcal{N}\left(0, \begin{bmatrix} \mathbf{A}\mathbf{K}\mathbf{A}^\top + \mathbf{\Omega}^{-1} & \mathbf{A}\mathbf{K}_* \\ (\mathbf{A}\mathbf{K}_*)^\top & \mathbf{K}_{**} \end{bmatrix}\right), \tag{43}$$

where $\mathbf{K}_*$ is the $NC \times C$ block diagonal matrix with blocks $K_\theta(\mathbf{X}, \mathbf{x}_*)$ and $\mathbf{K}_{**}$ is the $C \times C$ diagonal matrix with diagonal entries $k_\theta(\mathbf{x}_*, \mathbf{x}_*)$. The predictive distribution becomes:

$$p(\mathbf{f}_*|\mathbf{x}_*, \mathbf{X}, \mathbf{Y}, \boldsymbol{\omega}) = \mathcal{N}(\mathbf{f}_*|\boldsymbol{\mu}_*, \mathbf{\Sigma}_*), \text{ where}$$
$$\boldsymbol{\mu}_* = (\mathbf{A}\mathbf{K}_*)^\top(\mathbf{A}\mathbf{K}\mathbf{A}^\top + \mathbf{\Omega}^{-1})^{-1}\mathbf{\Omega}^{-1}\boldsymbol{\kappa} \text{ and} \tag{44}$$
$$\mathbf{\Sigma}_* = \mathbf{K}_{**} - (\mathbf{A}\mathbf{K}_*)^\top(\mathbf{A}\mathbf{K}\mathbf{A}^\top + \mathbf{\Omega}^{-1})^{-1}\mathbf{A}\mathbf{K}_*.$$

With $p(\mathbf{f}_*|\mathbf{x}_*, \mathbf{X}, \mathbf{Y}, \boldsymbol{\omega})$ in hand, the integral in (42) can easily be computed numerically for each class $c$ by forming the corresponding OVE linear transformation matrix $\mathbf{A}^c$ and then performing 1D Gaussian-Hermite quadrature on each dimension of $\mathcal{N}(\boldsymbol{\psi}_*^c|\mathbf{A}^c\boldsymbol{\mu}^*, \mathbf{A}^c\mathbf{\Sigma}_*\mathbf{A}^{c\top})$.

## F    DETAILED COMPARISON OF LIKELIHOODS

In this section we seek to better understand the behaviors of the softmax, OVE, logistic softmax, and Gaussian likelihoods for classification. For convenience, we summarize the forms of these likelihoods in Table 3.

Table 3: Likelihoods used in Section F.

| Likelihood | $\mathcal{L}(\mathbf{f} \,|\, y = c)$ |
|---|---|
| Softmax | $\dfrac{\exp(f_c)}{\sum_{c'} \exp(f_{c'})}$ |
| Gaussian | $\prod_{c'} \mathcal{N}(2 \cdot \mathbb{1}[c' = c] - 1 \,|\, \mu = f_{c'}, \sigma^2 = 1)$ |
| Logistic Softmax (LSM) | $\dfrac{\sigma(f_c)}{\sum_{c'} \sigma(f_{c'})}$ |
| One-vs-Each (OVE) | $\prod_{c' \neq c} \sigma(f_c - f_{c'})$ |

### F.1    HISTOGRAM OF CONFIDENCES

We sampled logits from $f_c \sim \mathcal{N}(0, 1)$ and plotted a histogram and kernel density estimate of the maximum output probability $\max_c p(y = c \,|\, \mathbf{f})$ for each of the likelihoods shown in Table 3, where $C = 5$. The results are shown in Figure 4. Logistic softmax is *a priori* underconfident: it puts little probability mass on confidence above 0.4. This may be due to the use of the sigmoid function which squashes large values of $f$. Gaussian likelihood and OVE are *a priori* overconfident in that they put a large amount of probability mass on confident outputs. Note that this is not a complete explanation, because GP hyperparameters such as the prior mean or Gaussian likelihood variance may be able to compensate for these imperfections to some degree. Indeed, we found it helpful to learn a constant mean for the logistic softmax likelihood, as mentioned in Section G.2.

### F.2    LIKELIHOOD VISUALIZATION

In order to visualize the various likelihoods under consideration, we consider a trivial classification task with a single observed example. We assume that there are three classes ($C = 3$) and the single example belongs to the first class ($y = 1$). We place the following prior on $\mathbf{f} = (f_1, f_2, f_3)^\top$:

$$p(\mathbf{f}) = \mathcal{N}\left(\mathbf{f} \,\middle|\, \boldsymbol{\mu} = \begin{bmatrix} 0 \\ 0 \\ 0 \end{bmatrix}, \mathbf{\Sigma} = \begin{bmatrix} 1 & 0 & 0 \\ 0 & 1 & 0 \\ 0 & 0 & 0 \end{bmatrix}\right). \tag{45}$$

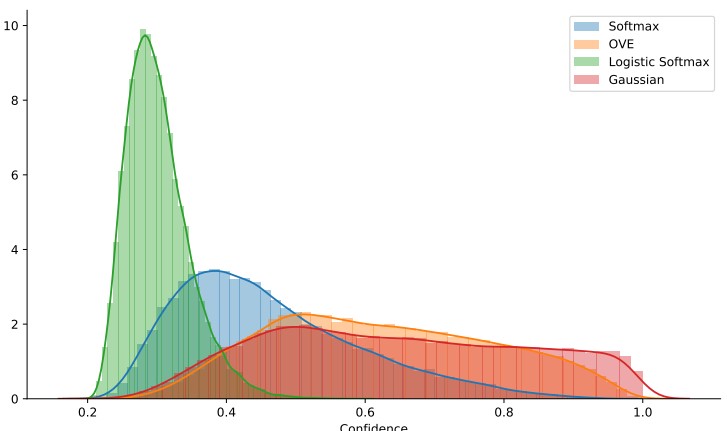

Figure 4: Histogram and kernel density estimate of confidence for randomly generated function samples $f_c \sim \mathcal{N}(0, 1)$. Normalized output probabilities were computed for $C = 5$ and a histogram of $\max_c p(y = c | \mathbf{f})$ was computed for 50,000 randomly generated simulations.

In other words, the prior for $f_1$ and $f_2$ is a standard normal and $f_3$ is clamped at zero (for ease of visualization). The likelihoods are plotted in Figure 5 and the corresponding posteriors are plotted in Figure 6.

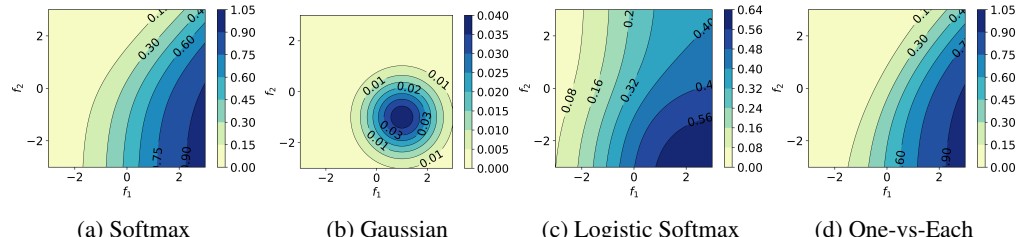

|                    |                     |                         |                        |
| ------------------ | ------------------- | ----------------------- | ---------------------- |
| (a) Softmax        | (b) Gaussian        | (c) Logistic Softmax    | (d) One-vs-Each        |

Figure 5: Plot of $\mathcal{L}(\mathbf{f} \mid y = 1)$, where $f_3$ is clamped to 0. The Gaussian likelihood penalizes configurations far away from $(f_1, f_2) = (1, -1)$. Logistic softmax is much flatter compared to softmax and has visibly different contours. One-vs-Each is visually similar to the softmax but penalizes $(f_1, f_2)$ near the origin slightly more.

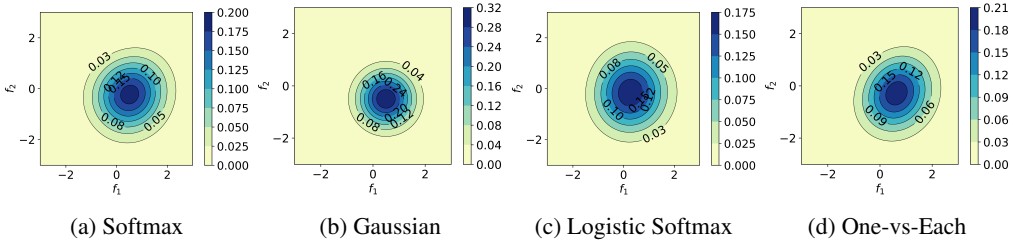

|                    |                     |                         |                        |
| ------------------ | ------------------- | ----------------------- | ---------------------- |
| (a) Softmax        | (b) Gaussian        | (c) Logistic Softmax    | (d) One-vs-Each        |

Figure 6: Plot of posterior $p(\mathbf{f} \mid y = 1)$, where $f_3$ is clamped to 0. The mode of each posterior distribution is similar, but each differs slightly in shape. Gaussian is more peaked about its mode, while logistic softmax is more spread out. One-vs-Each is similar to softmax, but is slightly more elliptical.

### F.3    2D IRIS EXPERIMENTS

We also conducted experiments on a 2D version of the Iris dataset (Fisher, 1936), which contains 150 examples across 3 classes. The first two features of the dataset were retained (sepal length and width). We used a zero-mean GP prior and an RBF kernel $k(\mathbf{x}, \mathbf{x}') = \exp\left(-\frac{1}{2}d(\mathbf{x}, \mathbf{x}')^2\right)$, where $d(\cdot, \cdot)$ is Euclidean distance. We considered training set sizes with 1, 2, 3, 4, 5, 10, 15, 20, 25, and 30 examples per class. For each training set size, we performed GP inference on 200 randomly generated train/test splits and compared the predictions across Gaussian, logistic softmax, and one-vs-each likelihoods.

Predictions at a test point $\mathbf{x}_*$ were made by applying the (normalized) likelihood to the posterior predictive mean $\bar{\mathbf{f}}_*$. The predictive probabilities for each likelihood is shown in Figure 7 for a randomly generated train/test split with 30 examples per class. Test predictive accuracy, Brier score, expected calibration error, and evidence lower bound (ELBO) results across various training set sizes are shown in Figure 8.

The ELBO is computed by treating each likelihood's posterior $q(\mathbf{f}|\mathbf{X}, \mathbf{Y})$ as an approximation to the softmax posterior $p(\mathbf{f}|\mathbf{X}, \mathbf{Y})$.

$$\begin{aligned}\text{ELBO}(q) &= \mathbb{E}_q[\log p(\mathbf{f}|\mathbf{X})] + \mathbb{E}_q[\log p(\mathbf{Y}|\mathbf{f})] - \mathbb{E}_q[\log q(\mathbf{f}|\mathbf{X}, \mathbf{Y})]\\ &= \log p(\mathbf{x}) - \text{KL}(q(\mathbf{f}|\mathbf{X}, \mathbf{Y})||p(\mathbf{f}|\mathbf{X}, \mathbf{Y})).\end{aligned}$$

Even though direct computation of the softmax posterior $p(\mathbf{f}|\mathbf{X}, \mathbf{y})$ is intractable, computing the ELBO is tractable. A larger ELBO indicates a lower KL divergence to the softmax posterior.

One-vs-Each performs well for accuracy, Brier score, and ELBO across the training set sizes. Gaussian performs best on expected calibration error through 15 examples per class, beyond which one-vs-each is better.

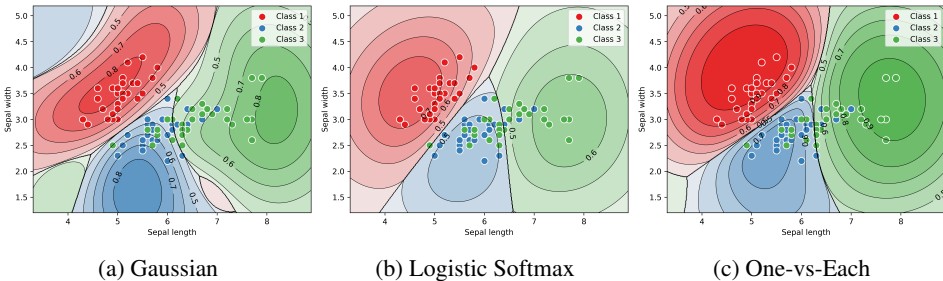

|  (a) Gaussian  |  (b) Logistic Softmax  |  (c) One-vs-Each  |

Figure 7: Training points (colored points) and maximum predictive probability for various likelihoods on the Iris dataset. The Gaussian likelihood produces more warped decision boundaries than the others. Logistic softmax tends to produce lower confidence predictions, while one-vs-each produces larger regions of greater confidence than the others.

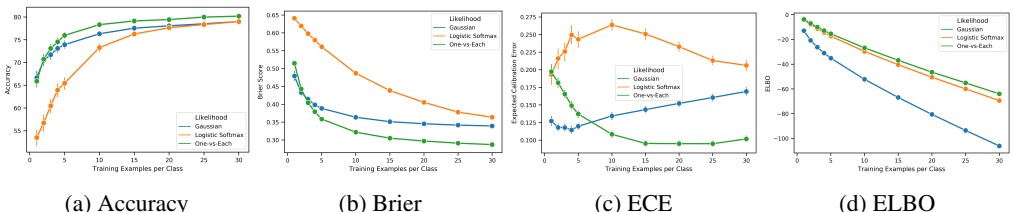

|  (a) Accuracy  |  (b) Brier  |  (c) ECE  |  (d) ELBO  |

Figure 8: Comparison across likelihoods in terms of test predictive accuracy, Brier score, expected calibration error (computed with 10 bins), and ELBO. Results are averaged over 200 randomly generated splits for each training set size (1, 2, 3, 4, 5, 10, 15, 20, 25, and 30 examples per class). Error bars indicate 95% confidence intervals.

## G  FEW-SHOT EXPERIMENTAL DETAILS

Here we provide more details about our experimental setup for our few-shot classification experiments, which are based on the protocol of (Patacchiola et al., 2020).

### G.1  DATASETS

We used the four dataset scenarios described below. The first three are the same used by Chen et al. (2019) and the final was proposed by Patacchiola et al. (2020).

- **CUB.** Caltech-UCSD Birds (CUB) (Wah et al., 2011) consists of 200 classes and 11,788 images. A split of 100 training, 50 validation, and 50 test classes was used (Hilliard et al., 2018; Chen et al., 2019).

- **mini-Imagenet.** The mini-Imagenet dataset (Vinyals et al., 2016) consists of 100 classes with 600 images per class. We used the split proposed by Ravi & Larochelle (2017), which has 64 classes for training, 16 for validation, and 20 for test.

- **mini-Imagenet→CUB.** This cross-domain transfer scenario takes the training split of mini-Imagenet and the validation & test splits of CUB.

- **Omniglot → EMNIST.** We use the same setup as proposed by Patacchiola et al. (2020). Omniglot (Lake et al., 2011) consists of 1,623 classes, each with 20 examples, and is augmented by rotations of 90 degrees to create 6,492 classes, of which 4,114 are used for training. The EMNIST dataset (Cohen et al., 2017), consisting of 62 classes, is split into 31 training and 31 test classes.

### G.2  FEW-SHOT CLASSIFICATION BASELINES

Here we explain the few-shot baselines in greater detail.

- **Feature Transfer** (Chen et al., 2019) involves first training an off-line classifier on the training classes and then training a new classification layer on the episode.

- **Baseline++** (Chen et al., 2019) is similar to Feature Transfer except it uses a cosine distance module prior to the softmax during fine-tuning.

- **Matching Networks** (Vinyals et al., 2016) can be viewed as a soft form of $k$-nearest neighbors that computes attention and sums over the support examples to form a predictive distribution over classes.

- **Prototypical Networks** (Snell et al., 2017) computes class means (prototypes) and forms a predictive distribution based on Euclidean distance to the prototypes. It can be viewed as a Gaussian classifier operating in an embedding space.

- **MAML** (Finn et al., 2017) performs one or a few steps of gradient descent on the support set and then makes predictions on the query set, backpropagating through the gradient descent procedure. For this baseline, we simply quote the classification accuracy reported by (Patacchiola et al., 2020).

- **RelationNet** (Sung et al., 2018) rather than using a predefined distance metric as in Matching Networks or Prototypical Networks instead learns a deep distance metric as the output of a neural network that accepts as input the latent representation of both examples. It is trained to minimize squared error of output predictions.

- **Deep Kernel Transfer (DKT)** (Patacchiola et al., 2020) relies on least squares classification (Rifkin & Klautau, 2004) to maintain tractability of Gaussian process posterior inference. In DKT, a separate binary classification task is formed for each class in one-vs-rest fashion by treating labels in $\{-1, +1\}$ as continuous targets. We include the results of DKT with the cosine kernel as implemented by Patacchiola et al. (2020), which is parameterized slightly differently from the version we used in (47):

$$k_{\text{dkt}}^{\cos}(\mathbf{x}, \mathbf{x}'; \boldsymbol{\theta}, \alpha, \nu) = \text{softplus}(\alpha) \cdot \text{softplus}(\nu) \cdot \frac{g_{\boldsymbol{\theta}}(\mathbf{x})^{\top} g_{\boldsymbol{\theta}}(\mathbf{x}')}{\|g_{\boldsymbol{\theta}}(\mathbf{x})\| \|g_{\boldsymbol{\theta}}(\mathbf{x}')\|}. \tag{46}$$

- **Bayesian MAML** (Yoon et al., 2018) relies on Stein Variational Gradient Descent (SVGD) (Liu & Wang, 2016) to get an approximate posterior distribution in weight-space. We compare to both the non-chaser version, which optimizes cross-entropy of query predictions, and the chaser version, which optimizes mean squared error between the approximate posterior on the support set and the approximate posterior on the merged support & query set. The non-chaser version is therefore related to predictive likelihood methods and the chaser version is more analogous to the marginal likelihood methods. For the non-chaser version, we used 20 particles and 1 step of adaptation at both train and test time. For the chaser version, we also used 20 particles. At train time, the chaser took 1 step and the leader 1 additional step. At test time, we used 5 steps of adaptation. Due to the slow performance of this method, we followed the advice of Yoon et al. (2018) and only performed adaptation on the final layer of weights, which may help explain the drop in performance relative to MAML. The authors released Tensorflow code for regression only, so we reimplemented this baseline for classification in PyTorch.

- **Amortized Bayesian Meta-Learning (ABML)** (Ravi & Beatson, 2019) performs a few steps of Bayes-by-backprop (Blundell et al., 2015) in order to infer a fully-factorized approximate posterior over the weights. The authors did not release code and so we implemented our own version of ABML in PyTorch. We found the weighting on the inner and outer KL divergences to be important for achieving good performance. We took the negative log likelihood to be mean cross entropy and used an inner KL weight of 0.01 and an outer KL weight of 0.001. These values were arrived upon by doing a small amount of hyperparameter tuning on the Omniglot$\rightarrow$ EMNIST dataset. We used $\alpha = 1.0$ and $\beta = 0.01$ for the Gamma prior over the weights. We only applied ABML to the weights of the network; the biases were learned as point estimates. We used 4 steps of adaptation and took 5 samples when computing expectations (using any more than this did not fit into GPU memory). We used the local reparameterization trick (Kingma et al., 2015) and flipout (Wen et al., 2018) when computing expectations in order to reduce variance. In order to match the architecture used by Ravi & Beatson (2019), we trained this baseline with 32 filters throughout the classification network. We trained each 1-shot ABML model for 800 epochs and each 5-shot ABML model for 600 epochs as the learning had not converged within the epoch limits specified in Section G.3.

- **Logistic Softmax GP** (Galy-Fajou et al., 2020) is the multi-class Gaussian process classification method that relies on the logistic softmax likelihood. Galy-Fajou et al. (2020) did not consider few-shot, but we use the same objectives described in Section 4.4 to adapt this method to FSC. In addition, we used the cosine kernel (see Section H for a description) that we found to work best with our OVE PG GPs. For this method, we found it important to learn a constant mean function (rather than a zero mean) in order to improve calibration.

### G.3 TRAINING DETAILS

All methods employed the commonly-used Conv4 architecture (Vinyals et al., 2016) (see Table 4 for a detailed specification), except ABML which used 32 filters throughout. All of our experiments used the Adam (Kingma & Ba, 2015) optimizer with learning rate $10^{-3}$. During training, all models used epochs consisting of 100 randomly sampled episodes. A single gradient descent step on the encoder network and relevant hyperparameters is made per episode. All 1-shot models are trained for 600 epochs and 5-shot models are trained for 400 epochs, except for ABML which was trained for an extra 200 epochs. Each episode contained 5 classes (5-way) and 16 query examples. At test time, 15 query examples are used for each episode. Early stopping was performed by monitoring accuracy on the validation set. The validation set was not used for retraining.

We train both marginal likelihood and predictive likelihood versions of our models. For Pólya-Gamma sampling we use the PyPólyaGamma package[3]. During training, we use a single step of Gibbs ($T$=1). For evaluation, we run until $T = 50$. In both training and evaluation, we use $M = 20$ parallel Gibbs chains to reduce variance.

---

[3]https://github.com/slinderman/pypolyagamma

Table 4: Specification of Conv4 architecture. `Conv2d` layers are $3 \times 3$ with stride 1 and `same` padding. `MaxPool2d` layers are $2 \times 2$ with stride 2 and `valid` padding.

| Output Size | Layers |
|---|---|
| $1 \times 28 \times 28$ | Input image |
| $64 \times 14 \times 14$ | Conv2d |
| | BatchNorm2d |
| | ReLU |
| | MaxPool2d |
| $64 \times 7 \times 7$ | Conv2d |
| | BatchNorm2d |
| | ReLU |
| | MaxPool2d |
| $64 \times 3 \times 3$ | Conv2d |
| | BatchNorm2d |
| | ReLU |
| | MaxPool2d |
| $64 \times 1 \times 1$ | Conv2d |
| | BatchNorm2d |
| | ReLU |
| | MaxPool2d |
| 64 | Flatten |

(a) Omniglot→EMNIST dataset.

| Output Size | Layers |
|---|---|
| $3 \times 84 \times 84$ | Input image |
| $64 \times 42 \times 42$ | Conv2d |
| | BatchNorm2d |
| | ReLU |
| | MaxPool2d |
| $64 \times 21 \times 21$ | Conv2d |
| | BatchNorm2d |
| | ReLU |
| | MaxPool2d |
| $64 \times 10 \times 10$ | Conv2d |
| | BatchNorm2d |
| | ReLU |
| | MaxPool2d |
| $64 \times 5 \times 5$ | Conv2d |
| | BatchNorm2d |
| | ReLU |
| | MaxPool2d |
| 1600 | Flatten |

(b) All other datasets.

## H    EFFECT OF KERNEL CHOICE ON CLASSIFICATION ACCURACY

In this section, we examine the effect of kernel choice on classification accuracy for our proposed One-vs-Each Pólya-Gamma OVE GPs.

**Cosine Kernel.**    In the main paper, we showed results for the following kernel, which we refer to as the "cosine" kernel due to its resemblance to cosine similarity:

$$k^{\cos}(\mathbf{x}, \mathbf{x}'; \boldsymbol{\theta}, \alpha) = \exp(\alpha) \frac{g_{\boldsymbol{\theta}}(\mathbf{x})^{\top} g_{\boldsymbol{\theta}}(\mathbf{x}')}{\|g_{\boldsymbol{\theta}}(\mathbf{x})\| \|g_{\boldsymbol{\theta}}(\mathbf{x}')\|}, \tag{47}$$

where $g_{\boldsymbol{\theta}}(\cdot)$ is a deep neural network that outputs a fixed-dimensional encoded representation of the input and $\alpha$ is the scalar log output scale. Both $\boldsymbol{\theta}$ and $\alpha$ are considered hyperparameters and learned simultaneously as shown in Algorithm 1. We found that this kernel works well for a range of datasets and shot settings. We note that the use of cosine similarity is reminiscent of the approach taken by Baseline++ method of (Chen et al., 2019), which computes the softmax over cosine similarity to class weights.

Here we consider three additional kernels: linear, RBF, and normalized RBF.

**Linear Kernel.**    The linear kernel is defined as follows:

$$k^{\lin}(\mathbf{x}, \mathbf{x}'; \boldsymbol{\theta}, \alpha) = \frac{1}{D} \exp(\alpha) g_{\boldsymbol{\theta}}(\mathbf{x})^{\top} g_{\boldsymbol{\theta}}(\mathbf{x}'), \tag{48}$$

where $D$ is the output dimensionality of $g_{\boldsymbol{\theta}}(\mathbf{x})$. We apply this dimensionality scaling because the dot product between $g_{\boldsymbol{\theta}}(\mathbf{x})$ and $g_{\boldsymbol{\theta}}(\mathbf{x}')$ may be large depending on $D$.

**RBF Kernel.**    The RBF (also known as squared exponential) kernel can be defined as follows:

$$k^{\rbf}(\mathbf{x}, \mathbf{x}'; \boldsymbol{\theta}, \alpha, \ell) = \exp(\alpha) \exp\left(-\frac{1}{2D \exp(\ell)^2} \|g_{\boldsymbol{\theta}}(\mathbf{x}) - g_{\boldsymbol{\theta}}(\mathbf{x}')\|^2\right), \tag{49}$$

where $\ell$ is the log lengthscale parameter (as with $\alpha$, we learn the $\ell$ alongside $\boldsymbol{\theta}$).

**Normalized RBF Kernel.** Finally, we consider a normalized RBF kernel similar in spirit to the cosine kernel:

$$k^{\text{rbf-norm}}(\mathbf{x}, \mathbf{x}'; \boldsymbol{\theta}, \alpha, \ell) = \exp(\alpha) \exp\left(-\frac{1}{2\exp(\ell)^2} \left\| \frac{g_{\boldsymbol{\theta}}(\mathbf{x})}{\|g_{\boldsymbol{\theta}}(\mathbf{x})\|} - \frac{g_{\boldsymbol{\theta}}(\mathbf{x}')}{\|g_{\boldsymbol{\theta}}(\mathbf{x}')\|} \right\|^2 \right). \qquad (50)$$

The results of our Pólya-Gamma OVE GPs with different kernels can be found in Tables 5 and 6. In general, we find that the cosine kernel works best overall, with the exception of Omniglot→EMNIST, where RBF does best.

Table 5: Classification accuracy for Pólya-Gamma OVE GPs (our method) using different kernels. Cosine is overall the best, followed closely by linear. RBF-based kernels perform worse, except for the Omniglot→EMNIST dataset. Evaluation is performed on 5 randomly generated sets of 600 test episodes. Standard deviation of the mean accuracy is also shown. ML = Marginal Likelihood, PL = Predictive Likelihood.

| | | CUB | | mini-ImageNet | |
|---|---|---|---|---|---|
| Kernel | Objective | 1-shot | 5-shot | 1-shot | 5-shot |
| Cosine | ML | **63.98 ± 0.43** | 77.44 ± 0.18 | 50.02 ± 0.35 | 64.58 ± 0.31 |
| Linear | ML | 62.48 ± 0.26 | 77.94 ± 0.21 | **50.81 ± 0.30** | 66.66 ± 0.45 |
| RBF | ML | 58.49 ± 0.40 | 75.50 ± 0.18 | 50.33 ± 0.26 | 64.62 ± 0.37 |
| RBF (normalized) | ML | 62.75 ± 0.32 | 78.71 ± 0.08 | 50.26 ± 0.31 | 64.84 ± 0.39 |
| Cosine | PL | 60.11 ± 0.26 | **79.07 ± 0.05** | 48.00 ± 0.24 | **67.14 ± 0.23** |
| Linear | PL | 60.44 ± 0.39 | 78.54 ± 0.19 | 47.29 ± 0.31 | 66.66 ± 0.36 |
| RBF | PL | 56.18 ± 0.69 | 77.96 ± 0.19 | 48.06 ± 0.28 | 66.66 ± 0.39 |
| RBF (normalized) | PL | 59.78 ± 0.34 | 78.42 ± 0.13 | 47.51 ± 0.20 | 66.42 ± 0.36 |

Table 6: Cross-domain classification accuracy for Pólya-Gamma OVE GPs (our method) using different kernels. The experimental setup is the same as Table 5.

| | | Omniglot→EMNIST | | mini-ImageNet→CUB | |
|---|---|---|---|---|---|
| Kernel | Objective | 1-shot | 5-shot | 1-shot | 5-shot |
| Cosine | ML | 68.43 ± 0.67 | 86.22 ± 0.20 | **39.66 ± 0.18** | 55.71 ± 0.31 |
| Linear | ML | 72.42 ± 0.49 | 88.27 ± 0.20 | 39.61 ± 0.19 | 55.07 ± 0.29 |
| RBF | ML | **78.05 ± 0.38** | 88.98 ± 0.16 | 36.99 ± 0.07 | 51.75 ± 0.27 |
| RBF (normalized) | ML | 75.51 ± 0.47 | 88.86 ± 0.16 | 38.42 ± 0.16 | 54.20 ± 0.13 |
| Cosine | PL | 77.00 ± 0.50 | 87.52 ± 0.19 | 37.49 ± 0.11 | **57.23 ± 0.31** |
| Linear | PL | 75.87 ± 0.43 | 88.77 ± 0.10 | 36.83 ± 0.27 | 56.46 ± 0.22 |
| RBF | PL | 74.62 ± 0.35 | **89.87 ± 0.13** | 35.06 ± 0.25 | 55.12 ± 0.21 |
| RBF (normalized) | PL | 76.01 ± 0.31 | 89.42 ± 0.16 | 37.50 ± 0.28 | 56.80 ± 0.39 |

## I   ADDITIONAL CALIBRATION RESULTS

In Figure 9, we include calibration results for mini-Imagenet and Omniglot→EMNIST. They follow similar trends to the results presented in Section 5.2.

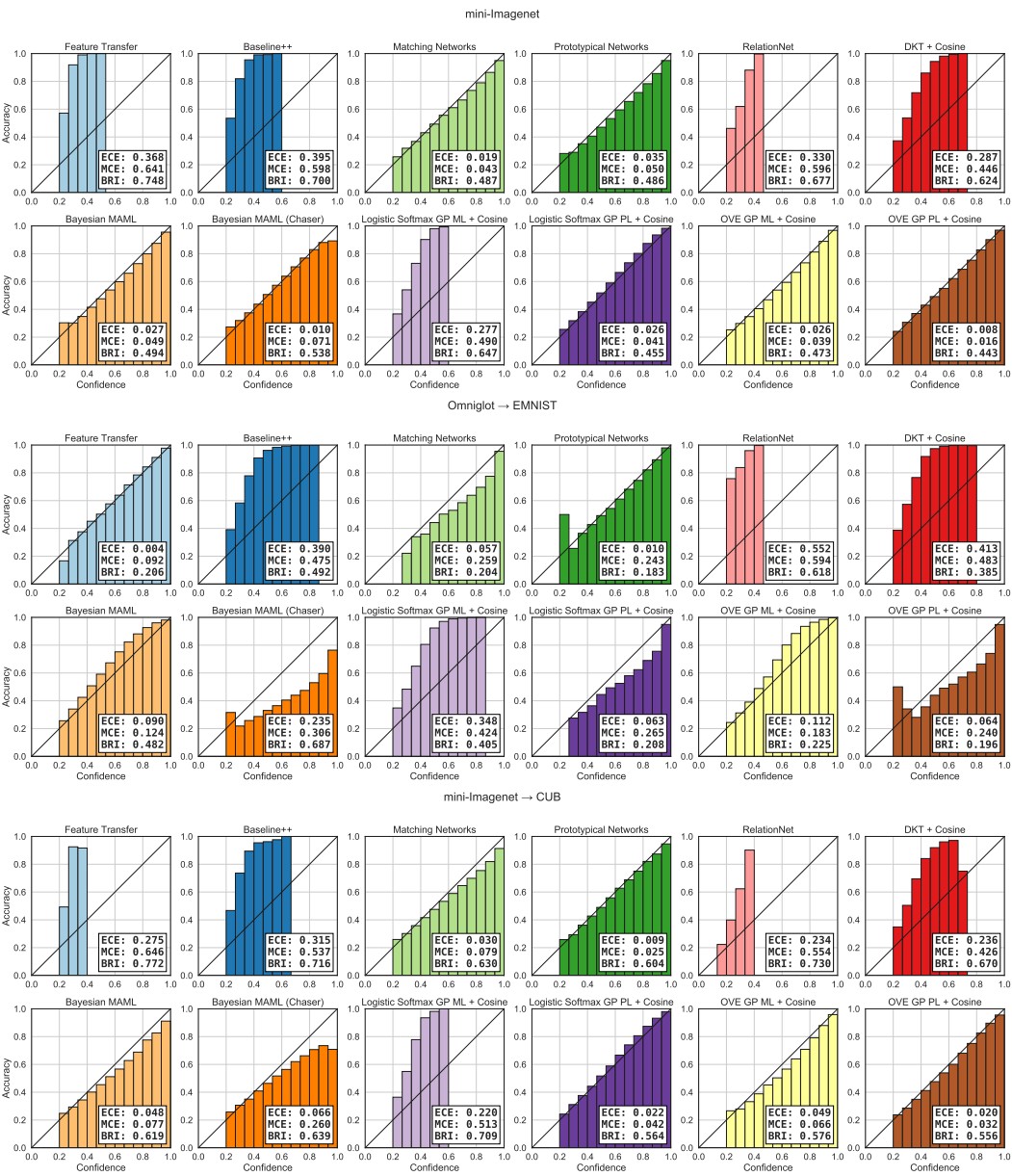

Figure 9: Reliability diagrams, expected calibration error, maximum calibration error, and Brier scores for 5-shot 5-way tasks on mini-Imagenet, Omniglot→EMNIST, and mini-Imagenet→CUB. Metrics are computed on 3,000 random tasks from the test set.

## J  QUANTITATIVE ROBUSTNESS TO INPUT NOISE RESULTS

In this section we include quantitative results for the robustness to input noise results presented in Figure 2. Results for Gaussian noise are shown in Table 7, impulse noise in Table 8, and defocus blur in Table 9.

Table 7: Accuracy (%) and Brier Score when applying Gaussian noise corruption of severity 5 to both the support and query set of test-time episodes. Results were evaluated across 1,000 randomly generated 5-shot 5-way tasks.

| Method | CUB | | mini-ImageNet | | mini-ImageNet→CUB | |
|---|---|---|---|---|---|---|
| | Acc. (↑) | Brier (↓) | Acc. (↑) | Brier (↓) | Acc. (↑) | Brier (↓) |
| Feature Transfer | 30.45 | 0.775 | 22.58 | 0.799 | 22.75 | 0.799 |
| Baseline++ | 22.60 | 0.798 | 23.82 | 0.797 | 24.13 | 0.797 |
| MatchingNet | 26.72 | 0.803 | 24.80 | 0.797 | 23.59 | 0.804 |
| ProtoNet | 32.28 | 0.778 | 29.97 | 0.781 | 32.30 | 0.779 |
| RelationNet | 25.23 | 0.799 | 23.69 | 0.800 | 20.00 | 0.800 |
| DKT + Cosine | 29.54 | 0.779 | 27.78 | 0.792 | 31.94 | 0.782 |
| Bayesian MAML | 22.79 | 0.905 | 20.52 | 0.963 | 20.46 | 0.949 |
| Bayesian MAML (Chaser) | 20.20 | 1.133 | 20.41 | 1.118 | 21.39 | 1.039 |
| LSM GP + Cosine (ML) | 27.92 | 0.787 | 22.43 | 0.798 | 22.36 | 0.799 |
| LSM GP + Cosine (PL) | 31.21 | 0.772 | 31.77 | 0.768 | **34.74** | **0.754** |
| OVE PG GP + Cosine (ML) [ours] | 32.27 | 0.774 | 29.99 | 0.776 | 29.97 | 0.784 |
| OVE PG GP + Cosine (PL) [ours] | **33.01** | **0.771** | **33.29** | **0.760** | 31.41 | 0.764 |

Table 8: Accuracy (%) and Brier Score when applying impulse noise corruption of severity 5 to both the support and query set of test-time episodes. Results were evaluated across 1,000 randomly generated 5-shot 5-way tasks.

| Method | CUB | | mini-ImageNet | | mini-ImageNet→CUB | |
|---|---|---|---|---|---|---|
| | Acc. (↑) | Brier (↓) | Acc. (↑) | Brier (↓) | Acc. (↑) | Brier (↓) |
| Feature Transfer | 30.20 | 0.776 | 23.54 | 0.798 | 22.87 | 0.799 |
| Baseline++ | 28.05 | 0.790 | 23.72 | 0.798 | 25.58 | 0.795 |
| MatchingNet | 28.25 | 0.790 | 23.80 | 0.803 | 23.21 | 0.811 |
| ProtoNet | 32.12 | 0.774 | 28.81 | 0.783 | 32.70 | 0.775 |
| RelationNet | 25.23 | 0.799 | 23.13 | 0.800 | 20.00 | 0.800 |
| DKT + Cosine | 29.74 | 0.778 | 29.11 | 0.789 | 32.26 | 0.781 |
| Bayesian MAML | 22.76 | 0.903 | 20.50 | 0.970 | 20.56 | 0.950 |
| Bayesian MAML (Chaser) | 20.25 | 1.172 | 20.51 | 1.116 | 21.45 | 1.022 |
| LSM GP + Cosine (ML) | 28.18 | 0.787 | 21.82 | 0.799 | 23.64 | 0.797 |
| LSM GP + Cosine (PL) | 32.10 | **0.769** | 30.22 | 0.776 | **35.09** | **0.751** |
| OVE PG GP + Cosine (ML) [ours] | 31.41 | 0.778 | 29.66 | 0.778 | 30.28 | 0.783 |
| OVE PG GP + Cosine (PL) [ours] | **33.36** | 0.772 | **33.23** | **0.761** | 32.06 | 0.762 |

Table 9: Accuracy (%) and Brier Score when applying defocus blur corruption of severity 5 to both the support and query set of test-time episodes. Results were evaluated across 1,000 randomly generated 5-shot 5-way tasks.

| Method | CUB | | mini-ImageNet | | mini-ImageNet→CUB | |
|---|---|---|---|---|---|---|
| | Acc. (↑) | Brier (↓) | Acc. (↑) | Brier (↓) | Acc. (↑) | Brier (↓) |
| **Feature Transfer** | 38.03 | 0.734 | 33.06 | 0.791 | 33.47 | 0.792 |
| **Baseline++** | 42.55 | 0.710 | 35.89 | 0.761 | 39.88 | 0.740 |
| **MatchingNet** | 44.43 | 0.682 | 34.43 | 0.754 | 35.95 | 0.741 |
| **ProtoNet** | 46.78 | **0.676** | 36.92 | **0.737** | 41.45 | 0.714 |
| **RelationNet** | 40.81 | 0.759 | 30.11 | 0.790 | 25.69 | 0.794 |
| **DKT + Cosine** | 45.34 | 0.695 | 38.29 | **0.737** | **45.17** | **0.703** |
| **Bayesian MAML** | 42.65 | 0.697 | 30.63 | 0.808 | 37.32 | 0.736 |
| **Bayesian MAML (Chaser)** | 40.66 | 0.881 | 29.93 | 1.121 | 31.33 | 1.125 |
| **LSM GP + Cosine (ML)** | 45.37 | 0.706 | 34.10 | 0.769 | 39.66 | 0.753 |
| **LSM GP + Cosine (PL)** | 48.55 | 0.690 | **39.46** | **0.737** | 43.15 | 0.714 |
| **OVE PG GP + Cosine (ML)** [ours] | 46.46 | 0.701 | 37.65 | 0.775 | 43.48 | 0.723 |
| **OVE PG GP + Cosine (PL)** [ours] | **49.44** | 0.695 | 38.95 | 0.780 | 43.66 | 0.720 |

