# OpenReview forum: "Bayesian Few-Shot Classification with One-vs-Each Pólya-Gamma Augmented Gaussian Processes"
_ICLR.cc/2021/Conference — ICLR 2021 Poster_

### Official Review · AnonReviewer2 · 2020-10-20
**review for #2193**

**Rating:** 6
**Confidence:** 4

**Review:**

This work studies the GP-based few-shot classification problem with one-vs-each softmax approximation and polya-gamma augmentation. It points out the existing problems of the current schemes: (1) the non-conjugacy of GP classification, which is solved by the augmentation of Polya-gamma random variables; (2) the incompatibility of Polya-gamma augmentation with softmax link function, which is addressed by the one-vs-each softmax approximation. The theoretical analysis is solid with only some minor typos. Also, a lot of experimental comparisons are conducted between the proposed model with alternatives w.r.t classification accuracy, uncertainty quantification, noise robustness and out-of-episode detection.

I recommend acceptance of the paper for the reasons below. Although the GP classification with Polya-gamma random variables is not new, the one-vs-each softmax approximation is a new idea as far as I know to reconcile the softmax link function with Polya-gamma augmentation. Another similar work is “Multi-class gaussian process classification made conjugate: Efficient inference via data augmentation, UAI 2019, providing a logistic-softmax likelihood to achieve the same effect, which is cited as LSM-GP by the work. Also, a large number of experiments are conducted and baselines are compared to validate the conclusion.

Weakness: A major concern is the work does not provide enough theoretical analysis about why the OVE likelihood is better than LSM. From the conjugacy perspective, both likelihood functions achieve the conjugate form. Besides, if my understanding is right, the LSM is an accurate “softmax” likelihood while the OVE is an approximation (lower-bound of the real softmax), why does the approximated likelihood (OVE) performs better than the accurate one (LSM) in all experiments? Is there any intuition behind this phenomenon? Although the author provided some evidence in experiments and sec.6 that OVE is better than LSM, some deep theoretical analysis is needed. Also, in almost all experiments, the author stated the proposed OVE “is one of the top performing methods”. I am not interested in such statement but the intuition analysis why it is better or worse. The work should not focus on SOTA-chasing but the delicate analytical solution, even if the performance is not the best.

Some minor concerns: In the comparison of likelihood in Fig.5, it is farfetched to stated the OVE is close to softmax, as it is obviously closer to the Gaussian likelihood which is overconfident as stated in the paper.
In sec4.2, it is unclear how to reduce the complexity from C^3N^3 to CN^3. It would be better to add a few words to describe it.
In sec4.3, how to marginalize over latent functions in both objectives? I did not find any details in the submission and appendix.
In sec5.3, if my understanding is right, the robustness to input noise comes from the GP smooth effect, so again why does the OVE-GP model outperform other GP-based models? Any intuition behind this?


Typo: sec4.1 \vec(f)\simGP(m,k)   f should be a function not vector
Eq(7) equal -> proportional or add normalization. The right hand side is not a pdf.

---

> ### Author Response · Authors · 2020-11-22
> **Thank you for your review**
>
> Thank you for your review and comments.
>
> An important question raised in your review (and in the others as well) is how our proposed one-versus-each (OVE) likelihood compares to other approximations to the standard softmax. In order to better understand the properties of our OVE likelihood relative to the Gaussian likelihood (used by GPNet [1]) and the logistic softmax likelihood (used by LSM GP [2]), we have conducted some simple experiments. Details can be found at this anonymized link: https://github.com/iclr2021-paper2193/paper2193-additional-experiments.
>
> The results for the likelihoods on a single example (Figures 1 and 2) show that the OVE likelihood is quite similar to the softmax and produces a similar posterior over f. Similarly, the results on the Iris dataset (Figures 3 and 4) shows that one-vs-each has better accuracy and calibration than the other likelihoods.
>
> *Why does the approximated likelihood (OVE) performs better than the accurate one (LSM) in all experiments? Is there any intuition behind this phenomenon?*
>
> Both OVE and LSM are alternatives to the softmax likelihood that admit tractable inference through data augmentation. OVE is a lower bound on the softmax likelihood, whereas LSM proposes an alternate form that replaces the exponentials of the softmax with sigmoids. It is difficult to conclusively state which of the two likelihoods is better, but we have attempted to provide some intuition in the linked document.
>
> In Figures 1 and 2 of the linked document, we have plotted the likelihoods as a function of f_1 and f_2, and OVE is visually quite similar to the softmax, with the main difference being that it is a bit flatter towards the origin.
>
> Even though OVE is a lower bound on the softmax, it appears to be a reasonably accurate bound. Empirically this tends to translate to better performance than LSM, which has a much different shape and form than the softmax.
>
> *In sec 4.2, it is unclear how to reduce the complexity from C^3N^3 to CN^3. It would be better to add a few words to describe it.*
>
> We will update the paper to include a more detailed description. The main idea is to avoid computation of the posterior covariance by first sampling f from the prior and then updating this f according to the procedure described by [3] to get a posterior sample.
>
> *In sec 4.3, how to marginalize over latent functions in both objectives? I did not find any details in the submission and appendix.*
>
> The marginal likelihood marginalizes over support function values f, and the predictive likelihood marginalizes over support function values $f$ and query function values $f*$. For predictive likelihood, please refer to Appendix D. For marginal likelihood, the marginalization is implicit in the $p(Y | X, \omega)$ expression in Appendix B (since $p(f|X)$ is Gaussian and $p(Y | f, \omega)$ is proportional a Gaussian), but we will expand to make this more clear.
>
> *In sec 5.3, if my understanding is right, the robustness to input noise comes from the GP smooth effect, so again why does the OVE-GP model outperform other GP-based models? Any intuition behind this?*
>
> The results indicate that GP-based models in general do relatively well on this task. We hypothesize this is due to their ability to marginalize over model uncertainty. As to why OVE outperforms the other GP-based models, please refer to the linked document.
>
> [1] Patacchiola et al. Deep kernel transfer in gaussian processes for few-shot learning. arXiv preprint arXiv:1910.05199 (2019).
>
> [2] Galy-Fajou et al. Multi-class gaussian process classification made conjugate: Efficient inference via data augmentation. UAI (2019).
>
> [3] Doucet. A Note on Efficient Conditional Simulation of Gaussian Distributions. (2010)

---

### Official Review · AnonReviewer3 · 2020-10-28
**Exellent paper on Bayesian modeling with Gaussian processes for important uncertainty management in Few-shot classificaiton without compromizing on accuracy.**

**Rating:** 8
**Confidence:** 3

**Review:**

Summary
--------
The authors considers the problem in Few-shot classification and addresses the need for uncertainty management (calibrated output uncertainty, robustness to input noise, and out-of-episode detection) while maintaining high accuracy. To this end the authors propose a novel approach based on Gaussian process classification with Polya-gamma augmentation, one-vs-each Softmax posterior approximation and with a novel cosine *similarity* kernel (in composition with deep kernels). The latent variables from the Poly-gamma augmentation and latent GP function are Gibbs sampled and GP hyperparameters (including the parameters of the NN making up the deep kernel) are optimized using gradient decent based on the samples. The approach is validated in comprehensive comparative empirical experiments involving multiple datasets, and is demonstrated to be top performing in both accuracy and uncertainty management.


Strong points
-------------
1. Well written paper, addressing an important problem in FSC with a well motivated and promising novel approach, filled with technical and methodology detail for completeness.
2. The approach combine high accuracy with calibrated output probabilities.
3. The performance is consistently strong at both robustness to input noise and out-of-episode detection.
4. The experiments are extensive w.r.t. the competitive approaches and have wide coverage given the different data sets used.

Weak points
-------------
Nothing obvious to me.

Reason for score
----------------
A well written paper, with several well motivated and empirically validated contributions, on an important topic. The paper seem to be technically correct and well placed in the literature. Especially the contributions on uncertainty quantification (both benchmarks and the proposed method) i believe are valuable and important for the FSC field, as well as for ICLR at large.

Minor comments
--------------
It would be better if Figure 2 can be made larger. Maybe by sharing the legend with Figure 3 and shortening one or two sentences slightly to make room?

---

> ### Author Response · Authors · 2020-11-22
> **Thank you for your review**
>
> Thank you for your review and comments. We hope that our work will lead to future work on studying uncertainty quantification for few-shot classification, which is an important yet understudied area.
>
> *It would be better if Figure 2 can be made larger. Maybe by sharing the legend with Figure 3 and shortening one or two sentences slightly to make room?*
>
> We agree that Figure 2 is a bit cramped. We will improve the legibility by focusing on a subset of these results in the main paper and putting the rest in the appendix.

---

### Official Review · AnonReviewer1 · 2020-10-28
**First Round Review by AnonReviewer1**

**Rating:** 7
**Confidence:** 4

**Review:**

1. Summary and contributions
This paper aims to improve the accuracy and uncertainty quantification in FSC using GP classifier. They use Polya-Gamma augmentation for tractable inference and introduce one-vs-each (OVE) approximation instead of softmax to apply PG’s property to the multi-class scenario.

2. Strengths
- Although using GP classifier with PG augmentation is not a novel idea, this is the first work tried in FSC. Also, OVE for handling multiple classes in PG augmentation is an interesting contribution.
- The method is clearly stated. It looks much more efficient than BNN-based algorithms.
- The authors demonstrated a range of experiments, including uncertainty quantification, noise robustness, and out-of-episode detection. Their OVE PG GP with cosine kernel consistently performs well in every experiment compared to other prevalent methods.

3. Weaknesses
- It is quite a novel idea to use PG augmentation with OVE approximation, but I think the authors are not well motivated about why their PG augmentation is necessary in FSC. If it were to quantify uncertainty, there are already GP-based algorithms as GPNet or LSM GP. Which property of PG augmentation makes it advantageous to other algorithms?
- In the experiments section, while the authors made an effort to implement extensive demonstrations, the discussion lacks. Specifically, while their OVE PG GP is consistently better than other GP-based classifiers, metric learning models, or Bayesian NN, the reason why their method outperforms the counterparts is still ambiguous. The authors should put more effort into explaining why OVE PG GP better classifies and captures uncertainty.
- Table 1 & Table 2. For OVE PG GP, it seems there is no clear winner between ML and PL objectives. For example, in CUB experiment, ML wins PL in 1-shot and PL wins ML in 5-shot. Some results have large deviations between them (e.g., nearly 9% difference in Omniglot->EMNIST 1-shot). Can you give a comment on the results of ML or PL: which is better to choose, and why they make inconsistent results in different experimental settings?
- sec 6. ‘The OVE likelihood is better suited to classification ~’: Is there a theoretical or intuitive ground on this claim? I understand that OVE likelihood is introduced due to PG’s incompatibility to multiple classification. However, is there a specific reason why OVE is better than LSM with respect to classification ability?
- In Appendix I, the authors compared several likelihood functions. It seems that OVE is the most similar to Gaussian likelihood. Then, why is OVE likelihood free of the ‘ill-suited nature of applying Gaussian likelihoods to the fundamentally discrete task of classification (sec 6)’?

4. Correctness
- sec 4.2. eq. (8) n→N, Y_(.c)→Y_(.c') Otherwise, entries of Af is not equal to f_i^(y_i )-f_i^c.

5. Additional questions or feedback
- sec 4.3. ‘We consider a zero-mean GP ~’: Can you explain why we should consider independent GPs for each class? Is it a common principle in GP classifier?
- Table 1. In mini-ImageNet 1-shot experiment, ABML result is extremely low and even lower than the original paper (37.65 in Table 1 vs. 45.0 in Ravi & Beatson, 2019.). Is there a significant change in the experimental setting?

6. Recommendation
This paper combines a novel idea of PG augmentation and OVE approximation into FSC, but still requires clear placement among the existing methods and more discussion on the reasoning of the results. I expect the authors to answer my concerns stated above during the rebuttal period. Thus, I vote this paper for rejecting (weak reject).

---

> ### Author Response · Authors · 2020-11-22
> **Thank you for your review**
>
> Thank you for your review and comments.
>
> An important question raised in your review (and in the others as well) is how our proposed one-versus-each (OVE) likelihood compares to other approximations to the standard softmax. In order to better understand the properties of our OVE likelihood relative to the Gaussian likelihood (used by GPNet [1]) and the logistic softmax likelihood (used by LSM GP [2]), we have conducted some simple experiments. Details can be found at this anonymized link: https://github.com/iclr2021-paper2193/paper2193-additional-experiments.
>
> The results for the likelihoods on a single example (Figures 1 and 2) show that the OVE likelihood is quite similar to the softmax and produces a similar posterior over f. Similarly, the results on the Iris dataset (Figures 3 and 4) shows that one-vs-each has better accuracy and calibration than the other likelihoods.
>
> *Which properties of our approach make it advantageous compared to other GP-based algorithms such as GPNet or LSM GP?*
>
> GPNet relies on least squares classification, which treats the labels as continuous targets in {-1, +1}. Classification can be performed simply enough by choosing the class with the greatest predicted f. However, as can be seen from the Iris experiments in the linked document, the decision boundaries are warped and the predictive probabilities are generally less well calibrated.
>
> LSM GP requires three augmentations (one gamma, one poisson, and one Pólya-gamma), compared to OVE which is simpler because it only requires one Pólya-gamma augmentation. We also found it important to learn a prior mean for LSM in our few-shot experiments. With a zero-mean prior, LSM performance drops 1-3% relative to the LSM results in the paper. This may be due to the squashing nature of sigmoid, which makes confident prediction difficult when the function values are close to 0.
>
> The one-vs-each likelihood behaves more similarly to a softmax GP than GPNet or LSM GP while still maintaining tractable inference. We believe this is the reason that its performance is better.
>
> *Which is better to choose, marginal likelihood or predictive likelihood?*
>
> To our knowledge, this is still an open question.
>
> Predictive likelihood computes the average probability of a query set label given the support set, whereas marginal likelihood computes the probability of observing all the labels (support and query) starting from scratch [3, Sec. 28.3]. If the number of shots at test-time will match the training shots, then we expect predictive likelihood to do better.
>
> Based on our results, marginal likelihood generally performed better for 1-shot than predictive likelihood. One possible reason might be that the posterior over f varies significantly in the 1-shot case, introducing increased stochasticity into the gradients that is not present for marginal likelihood, which computes the likelihood of the support and query set together.
>
> Additionally, predictive likelihood may be more robust to model misspecification [4; Sec. 5.4.2]. Thus when there is a bigger shift in the dataset, marginal likelihood may drop in performance more than predictive likelihood, which is what we observed in the Omniglot→EMNIST results.
>
> *In Appendix I, the authors compared several likelihood functions. It seems that OVE is the most similar to Gaussian likelihood.*
>
> Please refer to the linked document for a more detailed comparison of the properties of the likelihoods. While OVE and Gaussian both suffer from overconfidence and LSM suffers from underconfidence, of the 3 OVE produces results visually quite similar to the softmax.
>
> *Can you explain why we should consider independent GPs for each class? Is it a common principle in GP classifier?*
>
> Yes, to our knowledge this is common practice (see for example [4; Sec. 3.5]). Note that the GP posterior becomes coupled across classes for OVE and LSM due to the likelihood. GPNet on the other hand fits separate GPs for each class.
>
> *Sec 4.2. eq. (8)*
>
> Thank you, you are right and we will update this.
>
> *ABML result is extremely low and even lower than the original paper (37.65 in Table 1 vs. 45.0 in Ravi & Beatson, 2019.). Is there a significant change in the experimental setting?*
>
> The authors of ABML have not made code available, so the results in the paper are from our own implementation. The performance seems to be sensitive to the inner and outer KL weights but those hyperparameters were not listed in their paper. We have reached out to the authors for their code and will update the results provided we hear back from them.
>
> [1] Patacchiola et al. Deep kernel transfer in gaussian processes for few-shot learning. arXiv preprint arXiv:1910.05199 (2019).
>
> [2] Galy-Fajou et al. 2019. Multi-class gaussianprocess classification made conjugate: Efficient inference via data augmentation. UAI (2019).
>
> [3] MacKay. Information theory, inference and learning algorithms. (2003)
>
> [4] Rasmussen & Williams. Gaussian processes for machine learning. (2006)

---

### Official Review · AnonReviewer4 · 2020-10-28
**Review of "BAYESIAN FEW-SHOT CLASSIFICATION WITH ONE-VS-EACH POLYA-GAMMA AUGMENTED GAUSSIAN PROCESSES"**

**Rating:** 7
**Confidence:** 3

**Review:**


***
***
Update after author response:
The authors have addressed my comments and my recommendation remains unchanged.
***
***

##################################################

Summary:
The paper proposes a novel Bayesian method for few-shot classification. The proposed classifier makes use of the commonly-used Polya-gamma augmentation, but with likelihood replaced by a one-vs-each softmax approximation. The one-vs-each softmax approximation allows efficient computation in the posterior inference. The authors demonstrate better accuracy and uncertainty quantification in benchmark datasets.

##################################################

Pros:
1. The paper is nicely implemented and the proposed method is clearly motivated from existing methods and show promising performance.


##################################################

Cons:
1. While the robustness to input noise is evaluated, it is unclear why the proposed method can handle the situation where training and testing data contain different noise levels or generated from different mechanism.

2. It may be better to highlight more the challenges specifically in few-shot classification problems and motivate the OVE approximation in this context.

3. More justification of the the one-vs-each likelihood in a Bayesian setup is needed. It is unclear why it should be preferred except for computational reasons.

---

> ### Author Response · Authors · 2020-11-22
> **Thank you for your review**
>
> Thank you for your review and comments.
>
> *While the robustness to input noise is evaluated, it is unclear why the proposed method can handle the situation where training and testing data contain different noise levels or generated from different mechanism.*
>
> In our robustness to input noise experiments, both the support and query set are perturbed by input noise at test-time but not at train-time. When faced with input noise, we hypothesize that the learned deep kernel, while not perfect, will still manage to produce somewhat informative kernel similarities among similar input images. The resulting model uncertainty may be marginalized over by GP-based models such as ours.
>
> *It may be better to highlight more the challenges specifically in few-shot classification problems and motivate the OVE approximation in this context.*
>
> When there is little labeled data, as is the case for few-shot learning, there is a significant amount of model uncertainty. GPs are a way to capture and marginalize over that uncertainty. See the next point for why OVE would be preferable to other likelihoods.
>
> *More justification of the the one-vs-each likelihood in a Bayesian setup is needed. It is unclear why it should be preferred except for computational reasons.*
>
> Among GP-based methods, please refer to this document which compares the OVE likelihood to LSM and Gaussian: https://github.com/iclr2021-paper2193/paper2193-additional-experiments. The results indicate that OVE is overall more accurate.

---

### Author Response · Authors · 2020-11-25
**Updated Revision**

We have updated our submission to include the likelihood visualizations and comparison on Iris (Sections K and L). Please note in particular that we have recently added the ELBO as a metric for Iris in order to measure KL divergence to the softmax posterior (even though direct computation of the softmax posterior is intractable). A larger ELBO indicates a smaller KL divergence to the softmax posterior (since this KL is the gap between the ELBO and the softmax's marginal likelihood). Of the likelihoods under consideration, we found that One-vs-Each achieved the largest ELBO and thus lowest KL divergence to the softmax posterior.

We have also added Appendix E regarding efficiently sampling from the Gibbs conditional distribution over f.

---

### Decision · Program_Chairs · 2021-01-07
**Final Decision**

**Decision:**

Accept (Poster)

**Comment:**

The paper presents a Bayesian approach for classification able  to  adapt  to  novel  classes  given  only  a  few  labeled  examples. The models combines a one-vs-each approximation of the likelihood combined with a Gaussian process. This allows to resort to a data-augmentation scheme based on Polya-gamma random variables.
The paper is clearly written and combines existing techniques in a convincing manner; the experiments demonstrate better accuracy and uncertainty quantification on benchmark datasets.

I recommend acceptance.